# Bayesian Adaptive Calibration and Optimal Design

**Rafael Oliveira**\*
CSIRO's Data61
Sydney, Australia

**Dino Sejdinovic**
University of Adelaide
Adelaide, Australia

**David Howard**
CSIRO's Data61
Brisbane, Australia

**Edwin V. Bonilla**
CSIRO's Data61
Sydney, Australia

## Abstract

The process of calibrating computer models of natural phenomena is essential for applications in the physical sciences, where plenty of domain knowledge can be embedded into simulations and then calibrated against real observations. Current machine learning approaches, however, mostly rely on rerunning simulations over a fixed set of designs available in the observed data, potentially neglecting informative correlations across the design space and requiring a large amount of simulations. Instead, we consider the calibration process from the perspective of Bayesian adaptive experimental design and propose a data-efficient algorithm to run maximally informative simulations within a batch-sequential process. At each round, the algorithm jointly estimates the parameters of the posterior distribution and optimal designs by maximising a variational lower bound of the expected information gain. The simulator is modelled as a sample from a Gaussian process, which allows us to correlate simulations and observed data with the unknown calibration parameters. We show the benefits of our method when compared to related approaches across synthetic and real-data problems.

## 1 Introduction

In many scientific and engineering disciplines, computer simulation models form an essential part of the process of predicting and reasoning about complex phenomena, especially when real data is scarce. These simulation models depend on the inputs set by the user, commonly referred to as *designs*, and on a number of parameters representing unknown physical quantities, known as *calibration parameters*. The problem of setting these parameters so as to closely match observations of the real phenomenon is known as the calibration of computer models [1].

The seminal work by Kennedy and O'Hagan [1] introduces the Bayesian framework for calibration of simulation models, using Gaussian processes [2] to account for the differences between the model and reality, as well as for uncertainty in the calibration parameters. While the simulator is an essential tool when obtaining real data is expensive or unfeasible, each run of a simulator may itself involve significant computational resources, especially in applications such as climate science or complex engineering systems. In this situation, it is imperative to run simulations at carefully chosen settings of designs as well as of calibration inputs, using current knowledge to optimise resource use [3–5].

In this contribution, we bridge Bayesian calibration with adaptive experimental design [6] and use information-theoretic criteria [7] to guide the selection of simulation settings so that they are most informative about the true value of the calibration parameters. We refer to our approach as BACON (Bayesian Adaptive Calibration and Optimal desigN). BACON allows computational resources to be focused on simulations that provide the most value in terms of reducing epistemic uncertainty. Importantly, in contrast to prior work, it optimises designs *jointly* with calibration inputs in order to capture informative correlations across both spaces. Experimental results on synthetic experiments and a robotic gripper design problem demonstrate the benefits of BACON compared to competitive

---

\*Corresponding author: `rafael.dossantosdeoliveira@data61.csiro.au`

38th Conference on Neural Information Processing Systems (NeurIPS 2024).

baselines in terms of computational savings and the quality of the estimated posterior under similar computational constraints.

## 2 Problem formulation

Let $f : \mathcal{X} \to \mathcal{Y}$ represent a mapping of experimental designs $\mathbf{x} \in \mathcal{X}$ to the outcomes of a physical process $f(\mathbf{x}) \in \mathcal{Y} \subset \mathbb{R}$. We are given a set of observed outcomes $\mathbf{y}_R = [y_1, \ldots, y_R]^\mathsf{T}$ and their associated designs $\mathcal{X}_R := \{\mathbf{x}_i\}_{i=1}^R \subset \mathcal{X}$. Observations are corrupted by noise as $y_i = f(\mathbf{x}_i) + \nu_i$, where $\nu_i \sim \mathcal{N}(0, \sigma_\nu^2)$ is zero-mean Gaussian noise, for $i \in \{1, \ldots, R\}$. In addition, we have access to the output of a computer model $h : \mathcal{X} \times \Theta \to \mathbb{R}$ given a design input and simulation parameters. Given an optimal setting for the calibration parameters $\boldsymbol{\theta}^* \in \Theta$, the simulator $h(\mathbf{x}, \boldsymbol{\theta}^*)$, can be used to approximate the outcomes of the real physical process $f(\mathbf{x})$. However, $\boldsymbol{\theta}^*$ is unknown, and evaluations of the simulator $h$ are costly, though cheaper than executing real experiments evaluating $f$. Our task is to optimally estimate $\boldsymbol{\theta}^*$ given the real data $\mathbf{y}_R$, outputs of the simulator $h$ and a prior distribution $p(\boldsymbol{\theta}^*)$, representing initial assumptions about $\boldsymbol{\theta}^*$.

More concretely, let $\hat{\mathbf{y}}_S := [h(\hat{\mathbf{x}}_i, \hat{\boldsymbol{\theta}}_i)]_{i=1}^S$ represent simulated outcomes for a set of designs $\widehat{\mathcal{X}}_S := \{\hat{\mathbf{x}}_i\}_{i=1}^S \subset \mathcal{X}$ and simulation parameters $\widehat{\Theta}_S := \{\hat{\boldsymbol{\theta}}_i\}_{i=1}^S \subset \Theta$. Given the cost of running simulations, we will associate the simulator $h$ with a latent function (usually referred to as emulator) drawn from a Gaussian process (GP) prior and assume simulation outputs and real data follow a joint probability distribution $p(\mathbf{y}_R, \hat{\mathbf{y}}_S, \boldsymbol{\theta}^*)$.

In this setting, the Bayesian experimental design objective is to propose a sequence of simulations which will maximise the expected information gain (EIG) about $\boldsymbol{\theta}^*$:

$$
\begin{aligned}
\mathrm{EIG}(\widehat{\mathcal{X}}_S, \widehat{\Theta}_S) &:= \mathbb{H}(p(\boldsymbol{\theta}^*|\mathbf{y}_R)) - \mathbb{E}_{p(\hat{\mathbf{y}}_S|\widehat{\mathcal{X}}_S, \widehat{\Theta}_S, \mathbf{y}_R)}[\mathbb{H}(p(\boldsymbol{\theta}^*|\mathbf{y}_R, \hat{\mathbf{y}}_S))] \\
&= \mathbb{E}_{p(\hat{\mathbf{y}}_S|\widehat{\mathcal{X}}_S, \widehat{\Theta}_S, \mathbf{y}_R)}[\mathbb{D}_{\mathrm{KL}}(p(\boldsymbol{\theta}^*|\mathbf{y}_R, \hat{\mathbf{y}}_S)||p(\boldsymbol{\theta}^*|\mathbf{y}_R))] \\
&= \mathbb{I}(\boldsymbol{\theta}^*; \hat{\mathbf{y}}_S \mid \mathbf{y}_R, \widehat{\mathcal{X}}_S, \widehat{\Theta}_S),
\end{aligned}
\tag{1}
$$

where $\mathbb{H}(\cdot)$ represents the entropy of a probability distribution, $\mathbb{D}_{\mathrm{KL}}(\cdot||\cdot)$ denotes the Kullback-Leibler divergence, and $\mathbb{I}(\boldsymbol{\theta}^*; \hat{\mathbf{y}}_S \mid \mathbf{y}_R)$ is the mutual information between $\boldsymbol{\theta}^*$ and the simulator output $\hat{\mathbf{y}}_S$ given the real observations $\mathbf{y}_R$ and the simulator inputs to be optimized. We note here that, in our setting, the real observations $\mathbf{y}_R$ are always fixed. Therefore, intuitively, the EIG above captures the reduction in uncertainty that will be obtained when selecting $(\widehat{\mathcal{X}}_S, \widehat{\Theta}_S)$ averaged over all the possible outcomes $\hat{\mathbf{y}}_S$.

## 3 Related work

Our work consists of deriving a Bayesian adaptive experimental design (BAED) approach to the problem of calibration. Therefore, in the following, we will briefly discuss current literature on these two main research areas.

### 3.1 Adaptive experimental design

The problem of experimental design has a long history [8], spanning from classical fixed design patterns to modern adaptive approaches [9]. Optimal experimental design consists of selecting experiments which will maximise some form of criterion involving a measure of utility of the experiment and its associated costs [10]. Under the Bayesian formulation, uncertainty in the outcomes of the process is considered, and the optimality of a design is measured in terms of its expected utility [11]. Information theory then allows us to quantify information gain as a utility function, which is commonly applied in modern approaches to Bayesian experimental design [12].

The estimation of posterior distributions becomes a computational bottleneck for information-theoretic Bayesian frameworks. Recent work has focused on addressing the difficulties in estimating the expected information gain by means of, e.g., variational inference [13], density-ratio estimation [14], importance sampling [15], and the learning of efficient policies to propose designs [16, 17]. These methods, however, usually assume that the simulator is known and inexpensive to evaluate. In contrast, the simulations themselves are modelled as expensive experiments for us, and we apply

Gaussian process models as emulators to capture uncertainty over the black-box simulator. In addition, traditional BAED approaches assume that the prior is trivial to sample from and evaluate densities of, while in our case the starting prior is $p(\boldsymbol{\theta}^*|\mathbf{y}_R)$, which is likely non-trivial. We refer the reader to the recent review on modern Bayesian methods for experimental design by Rainforth et al. [18] for further details on BAED.

## 3.2 Active learning for calibration

Experimental design approaches generally aim towards the selection of designs for physical experiments, whereas we are concerned with the problem of running optimal simulated experiments for model calibration in the presence of real data. When simulations are resource-intensive, a few methods have been derived based on the Bayesian calibration framework proposed by Kennedy and O'Hagan [1]. Busby and Feraille [19] present an algorithm to learn GP emulators for a simulator which can then be combined with Bayesian inference algorithms, such as Markov chain Monte Carlo [20], to provide a posterior distribution over parameters. In their approach, the optimised variables are solely the calibration parameters, and the selection criterion is based on minimising the integrated mean-square error of the GP predictions. Many other approaches can be applied to this setting by modelling the simulator or its associated likelihood function as a GP, including Bayesian optimisation [3, 21, 22] and methods for adaptive Bayesian quadrature [23, 24]. Besides GPs, other algorithms focusing on the selection of calibration parameters have been derived using ensembles of neural networks [25] and deep reinforcement learning [26]. These frameworks, however, do not allow for the selection of simulation design points, usually keeping them co-located with the real data.

Allowing for design point decisions to be included, Leatherman et al. [4] presented approaches for combined simulation and physical experimental design following geometric and prediction-error-based criteria, though using an offline, non-sequential framework. More recently, Marmin and Filippone [5] derived a deep Gaussian process [27] framework for Bayesian calibration problems and discussed an application with experimental design among other examples. Their experimental design approach to calibration was based on choosing simulations that maximally reduce the variational posterior variance over the calibration parameters, as measured by the derivatives of the evidence lower bound with respect to (w.r.t.) variance parameters. In contrast, we aim to directly maximise the information gain w.r.t. the unknown calibration parameters.

## 4 Gaussian processes for Bayesian calibration

To estimate information gain, we need a probabilistic model which can correlate simulations with real data and the unknown parameters $\boldsymbol{\theta}^*$. Ideally, the model needs to allow for a computationally tractable conditioning on the parameters $\boldsymbol{\theta}^*$ and account for the discrepancy between real and simulated data. Hence, we follow the Bayesian calibration approach in Kennedy and O'Hagan [1] and model:

$$f(\mathbf{x}) = \rho h(\mathbf{x}, \boldsymbol{\theta}^*) + \varepsilon(\mathbf{x}), \quad \mathbf{x} \in \mathcal{X}, \quad \boldsymbol{\theta}^* \sim p(\boldsymbol{\theta}^*), \tag{2}$$

where $\varepsilon : \mathcal{X} \to \mathbb{R}$ represents the error (or discrepancy) between simulations and real outcomes, and $\rho \in \mathbb{R}$ accounts for possible differences in scale. We place Gaussian process priors on the simulator $h \sim \mathcal{GP}(0, \hat{k})$ and on the error function $\varepsilon \sim \mathcal{GP}(0, k_\varepsilon)$.

### 4.1 Bi-fidelity exact Gaussian process model

Since both $h$ and $\varepsilon$ are GPs, simulations and real outcomes can be jointly modelled as a single Gaussian process. In fact, both the simulator $h$ and the true function $f$ can be seen as different levels of fidelity of the same underlying process, with $h$ representing a coarser version of $f$. Namely, let $s \in \mathcal{S} := \{0, 1\}$ denote a fidelity parameter. The combined model is then given by:

$$\hat{f}(\mathbf{x}, \boldsymbol{\theta}, s) := \begin{cases} h(\mathbf{x}, \boldsymbol{\theta}), & s = 0 \\ \rho h(\mathbf{x}, \boldsymbol{\theta}) + \varepsilon(\mathbf{x}), & s = 1. \end{cases} \tag{3}$$

such that $f(\mathbf{x}) = \hat{f}(\mathbf{x}, \boldsymbol{\theta}^*, 1)$ and $h(\hat{\mathbf{x}}, \hat{\boldsymbol{\theta}}) = \hat{f}(\hat{\mathbf{x}}, \hat{\boldsymbol{\theta}}, 0)$, for any $\mathbf{x}, \hat{\mathbf{x}} \in \mathcal{X}$ and $\hat{\boldsymbol{\theta}} \in \Theta$. As a result, for arbitrary points in the joint space $\mathbf{z}, \mathbf{z}' \in \mathcal{Z} := \mathcal{X} \times \Theta \times \mathcal{S}$, the following covariance function parameterises the combined GP model $\hat{f} \sim \mathcal{GP}(0, k)$:

$$k(\mathbf{z}, \mathbf{z}') := k_\rho(s, s')\hat{k}((\mathbf{x}, \boldsymbol{\theta}), (\mathbf{x}', \boldsymbol{\theta}')) + ss'k_\varepsilon(\mathbf{x}, \mathbf{x}') \tag{4}$$

where $k_\rho(s, s') := (1 + s(\rho - 1))(1 + s'(\rho - 1))$, $\mathbf{z} := (\mathbf{x}, \boldsymbol{\theta}, s)$, and $\mathbf{z}' := (\mathbf{x}', \boldsymbol{\theta}', s')$. Therefore, any set of real and simulated evaluations are joint normally distributed under a combined GP model.

## 4.2   Joint probabilistic model and predictions

Let $\mathbf{Z}_R := \mathbf{Z}_R(\boldsymbol{\theta}^*) := [(\mathbf{x}_i, \boldsymbol{\theta}^*, 1)]_{i=1}^R$ represent the set of partially observed inputs for real data $\mathbf{y}_R$, and let $\widehat{\mathbf{Z}}_S := [(\hat{\mathbf{x}}_i, \hat{\boldsymbol{\theta}}, 0)]_{i=1}^S$ denote the current set of simulation inputs for the observations $\hat{\mathbf{y}}_S$. Under the GP prior, the joint probability model $p(\hat{\mathbf{y}}_S, \mathbf{y}_R, \boldsymbol{\theta}^*)$ can be decomposed as:

$$p(\hat{\mathbf{y}}_S, \mathbf{y}_R, \boldsymbol{\theta}^*) = p(\hat{\mathbf{y}}_S, \mathbf{y}_R | \boldsymbol{\theta}^*) p(\boldsymbol{\theta}^*) = \int_{\hat{\mathbf{f}}} p(\hat{\mathbf{y}}_S | \hat{\mathbf{f}}) p(\mathbf{y}_R | \hat{\mathbf{f}}, \boldsymbol{\theta}^*) p(\hat{\mathbf{f}} | \boldsymbol{\theta}^*) p(\boldsymbol{\theta}^*) \, \mathrm{d}\hat{\mathbf{f}}, \tag{5}$$

where $\hat{\mathbf{f}} := \hat{f}(\mathbf{Z}(\boldsymbol{\theta}^*)) \in \mathbb{R}^{R+S}$, and $\mathbf{Z}(\boldsymbol{\theta}^*) := \{\mathbf{Z}_R(\boldsymbol{\theta}^*), \widehat{\mathbf{Z}}_S\}$ corresponds to the full set of inputs. The GP prior then allows us to model real and simulated outcomes jointly as a Gaussian random vector $\hat{\mathbf{f}}$:

$$\hat{\mathbf{f}} | \boldsymbol{\theta}^* \sim \mathcal{N}(\mathbf{0}, \mathbf{K}(\boldsymbol{\theta}^*)), \tag{6}$$

where $\mathbf{K}(\boldsymbol{\theta}^*) := k(\mathbf{Z}(\boldsymbol{\theta}^*), \mathbf{Z}(\boldsymbol{\theta}^*)) = [k(\mathbf{z}, \mathbf{z}')]_{\mathbf{z}, \mathbf{z}' \in \mathbf{Z}(\boldsymbol{\theta}^*)}$ denotes the prior covariance matrix. Assuming a Gaussian noise model for the observations $y = f(\mathbf{x}, \boldsymbol{\theta}^*) + \varepsilon(\mathbf{x}) + \nu$, with $\nu \sim \mathcal{N}(0, \sigma_\nu^2)$, the marginal distribution over the observations $\mathbf{y} := [\mathbf{y}_R^\mathsf{T}, \hat{\mathbf{y}}_S^\mathsf{T}]^\mathsf{T}$ is available in closed form as:

$$p(\hat{\mathbf{y}}_S, \mathbf{y}_R | \boldsymbol{\theta}^*) = \mathcal{N}(\mathbf{y}; \mathbf{0}, \mathbf{K}(\boldsymbol{\theta}^*) + \boldsymbol{\Sigma}_\mathbf{y}), \tag{7}$$

where $\boldsymbol{\Sigma}_\mathbf{y}$ denotes the covariance matrix of the observation noise, i.e., $[\boldsymbol{\Sigma}_\mathbf{y}]_{ii} = \sigma_\nu^2$ for any $\mathbf{z}_i$ with $s_i = 1$, and $[\boldsymbol{\Sigma}_\mathbf{y}]_{ij} = 0$ elsewhere.[2]

Under the GP assumptions, we can make predictions about $\hat{y} = h(\hat{\mathbf{x}}, \hat{\boldsymbol{\theta}})$ at any pair of $\hat{\mathbf{x}}, \hat{\boldsymbol{\theta}} \in \mathcal{X} \times \Theta$. Conditioning on $\boldsymbol{\theta}^*$ and a dataset $\mathcal{D}_t := \{\mathcal{X}_R, \mathbf{y}_R, \widehat{\mathcal{X}}_t, \widehat{\Theta}_t, \hat{\mathbf{y}}_t\}$, let $\mathbf{Z}_t(\boldsymbol{\theta}^*) := \{\mathbf{Z}_R(\boldsymbol{\theta}^*), \widehat{\mathbf{Z}}_t\}$ denote the set of inputs up to time $t$ conditional on $\boldsymbol{\theta}^*$, and $\mathbf{y}_t$ the corresponding outputs. We then have that:

$$p(\hat{y} | \boldsymbol{\theta}^*, \hat{\mathbf{x}}, \hat{\boldsymbol{\theta}}, \mathcal{D}_t) = \mathcal{N}(\hat{y}; \mu_t(\hat{\mathbf{z}}; \boldsymbol{\theta}^*), \sigma_t^2(\hat{\mathbf{z}}; \boldsymbol{\theta}^*)), \tag{8}$$

for $\hat{\mathbf{z}} := (\hat{\mathbf{x}}, \hat{\boldsymbol{\theta}})$, where:

$$\mu_t(\hat{\mathbf{z}}; \boldsymbol{\theta}^*) := \mathbf{k}_t^\mathsf{T}(\hat{\mathbf{z}}; \boldsymbol{\theta}^*)^\mathsf{T} (\mathbf{K}_t(\boldsymbol{\theta}^*) + \boldsymbol{\Sigma}_{\mathbf{y}_t})^{-1} \mathbf{y}_t \tag{9}$$

$$k_t(\hat{\mathbf{z}}, \hat{\mathbf{z}}'; \boldsymbol{\theta}^*) := k(\hat{\mathbf{z}}, \hat{\mathbf{z}}') - \mathbf{k}_t(\hat{\mathbf{z}}; \boldsymbol{\theta}^*)^\mathsf{T} (\mathbf{K}_t(\boldsymbol{\theta}^*) + \boldsymbol{\Sigma}_{\mathbf{y}_t})^{-1} \mathbf{k}_t(\hat{\mathbf{z}}'; \boldsymbol{\theta}^*) \tag{10}$$

$$\sigma_t^2(\mathbf{z}; \boldsymbol{\theta}^*) := k_t(\hat{\mathbf{z}}, \hat{\mathbf{z}}; \boldsymbol{\theta}^*), \tag{11}$$

with $\mathbf{k}_t(\hat{\mathbf{z}}; \boldsymbol{\theta}^*) := k(\mathbf{Z}_t(\boldsymbol{\theta}^*), \hat{\mathbf{z}})$ and $\mathbf{K}_t(\boldsymbol{\theta}^*) := k(\mathbf{Z}_t(\boldsymbol{\theta}^*), \mathbf{Z}_t(\boldsymbol{\theta}^*))$. We next describe how to apply this model to derive a Bayesian adaptive calibration algorithm.

## 5   Bayesian adaptive calibration and optimal design

In this section, we describe an approach to design experiments for calibration of computer models that incorporates information gathered during the experiments iteratively. We refer to these types of designs as *adaptive*. Thus, we consider the sequential design of experiments setting, where at each iteration $t \in \mathbb{N}$, we optimise:

$$\begin{aligned}\mathrm{EIG}_t(\hat{\mathbf{x}}, \hat{\boldsymbol{\theta}}) &:= \mathbb{I}(\boldsymbol{\theta}^*; \hat{y} \mid \hat{\mathbf{x}}, \hat{\boldsymbol{\theta}}, \mathcal{D}_{t-1}) \\ &= \mathbb{H}(p(\boldsymbol{\theta}^* | \mathcal{D}_{t-1})) - \mathbb{E}_{\hat{y} \sim p(\hat{y} | \hat{\mathbf{x}}, \hat{\boldsymbol{\theta}}, \mathcal{D}_{t-1})}[\mathbb{H}(p(\boldsymbol{\theta}^* | \hat{y}, \hat{\mathbf{x}}, \hat{\boldsymbol{\theta}}, \mathcal{D}_{t-1}))] \\ &= \mathbb{E}_{p(\hat{y}, \boldsymbol{\theta}^* | \hat{\mathbf{x}}, \hat{\boldsymbol{\theta}}, \mathcal{D}_{t-1})}\left[\log \frac{p(\boldsymbol{\theta}^* | \hat{y}, \hat{\mathbf{x}}, \hat{\boldsymbol{\theta}}, \mathcal{D}_{t-1})}{p(\boldsymbol{\theta}^* | \mathcal{D}_{t-1})}\right],\end{aligned} \tag{12}$$

given the dataset $\mathcal{D}_{t-1} := \{\mathcal{X}_R, \mathbf{y}_R, \widehat{\mathcal{X}}_{t-1}, \widehat{\Theta}_{t-1}, \hat{\mathbf{y}}_{t-1}\}$ of observations. Given that the expected information gain is submodular [28], a sequential approach allows us to get close enough (usually a factor of at least $1 - 1/e$ [29]) to the optimal EIG over the whole experiment, while also allowing algorithmic decisions to adapt to current estimates for $p(\boldsymbol{\theta}^* | \mathcal{D}_t)$.

---

[2] In practice, we add a small *nugget* term to the diagonal of the noise covariance matrix for numerical stability.

In general, computing the full EIG objective (1), or its sequential version (12), is intractable, as that requires estimating the true posterior and its density conditioned on sampled data. Note that both $p(\boldsymbol{\theta}^*|\hat{y}, \hat{\mathbf{x}}, \hat{\boldsymbol{\theta}}, \mathcal{D}_{t-1})$ and $p(\hat{y}, \boldsymbol{\theta}^*|\hat{\mathbf{x}}, \hat{\boldsymbol{\theta}}, \mathcal{D}_{t-1})$ depend on the posterior $p(\boldsymbol{\theta}^*|\mathcal{D}_{t-1})$, as:

$$p(\boldsymbol{\theta}^*|\hat{y}, \hat{\mathbf{x}}, \hat{\boldsymbol{\theta}}, \mathcal{D}_{t-1}) = \frac{p(\hat{y}, \boldsymbol{\theta}^*|\hat{\mathbf{x}}, \hat{\boldsymbol{\theta}}, \mathcal{D}_{t-1})}{p(\hat{y}|\hat{\mathbf{x}}, \hat{\boldsymbol{\theta}}, \mathcal{D}_{t-1})} \tag{13}$$

$$p(\hat{y}, \boldsymbol{\theta}^*|\hat{\mathbf{x}}, \hat{\boldsymbol{\theta}}, \mathcal{D}_{t-1}) = p(\hat{y}|\boldsymbol{\theta}^*, \hat{\mathbf{x}}, \hat{\boldsymbol{\theta}}, \mathcal{D}_{t-1})p(\boldsymbol{\theta}^*|\mathcal{D}_{t-1}), \tag{14}$$

where the conditional predictive density $p(\hat{y}|\boldsymbol{\theta}^*, \hat{\mathbf{x}}, \hat{\boldsymbol{\theta}}, \mathcal{D}_{t-1})$ is Gaussian and available in closed form (Eq. 8). Clearly, in general, the true posterior is intractable, since $p(\boldsymbol{\theta}^*|\mathcal{D}_t) = \frac{p(\mathcal{D}_t|\boldsymbol{\theta}^*)p(\boldsymbol{\theta}^*)}{p(\mathcal{D}_t)}$ and $p(\mathcal{D}_t) = \int_{\Theta} p(\mathcal{D}_t|\boldsymbol{\theta}^*)p(\boldsymbol{\theta}^*) \, \mathrm{d}\boldsymbol{\theta}^*$ involves integration over the entire parameter space $\Theta$, which can be high dimensional and involve highly non-linear operations, such as computing inverse covariances. In addition, the marginal predictive density $p(\hat{y}|\hat{\mathbf{x}}, \hat{\boldsymbol{\theta}}, \mathcal{D}_{t-1}) = \int_{\Theta} p(\hat{y}, \boldsymbol{\theta}^*|\hat{\mathbf{x}}, \hat{\boldsymbol{\theta}}, \mathcal{D}_{t-1}) \, \mathrm{d}\boldsymbol{\theta}^*$ is also usually intractable for the same reasons.

## 5.1 Variational EIG lower bound

Following Foster et al. [13], we replace the EIG by a variational objective which does not require the true posterior density over $\boldsymbol{\theta}^*$. This formulation allows us to jointly estimate an approximation to the posterior and select optimal design points $\hat{\mathbf{x}}$ and simulation parameters $\hat{\boldsymbol{\theta}}$. Applying the variational lower bound by Barber and Agakov [30] to Eq. 12 yields the following alternative to the EIG:

$$\widehat{\mathrm{EIG}}_t(\hat{\mathbf{x}}, \hat{\boldsymbol{\theta}}, q) := \mathbb{E}_{p(\hat{y}, \boldsymbol{\theta}^*|\hat{\mathbf{x}}, \hat{\boldsymbol{\theta}}, \mathcal{D}_{t-1})}\left[\log \frac{q(\boldsymbol{\theta}^*|\hat{y}, \hat{\mathbf{x}}, \hat{\boldsymbol{\theta}})}{p(\boldsymbol{\theta}^*|\mathcal{D}_{t-1})}\right] \leq \mathrm{EIG}_t(\hat{\mathbf{x}}, \hat{\boldsymbol{\theta}}), \tag{15}$$

where $q(\boldsymbol{\theta}^*|\hat{y}, \hat{\mathbf{x}}, \hat{\boldsymbol{\theta}})$ is any conditional probability density model. The gap is given by the expected Kullback-Leibler (KL) divergence between the true and the variational posterior [13, Sec. A.1]:[3]

$$\mathrm{EIG}_t(\hat{\mathbf{x}}, \hat{\boldsymbol{\theta}}) - \widehat{\mathrm{EIG}}_t(\hat{\mathbf{x}}, \hat{\boldsymbol{\theta}}, q) = \mathbb{E}_{p(\hat{y}|\hat{\mathbf{x}}, \hat{\boldsymbol{\theta}}, \mathcal{D}_{t-1})}[\mathbb{D}_{\mathrm{KL}}(p(\boldsymbol{\theta}^*|\mathcal{D}_{t-1}, \hat{y})||q(\boldsymbol{\theta}^*|\hat{y}))] \geq 0. \tag{16}$$

Maximising the variational EIG lower bound w.r.t. the variational distribution $q$ then provides us with an approximation to $p(\boldsymbol{\theta}^*|\hat{y}, \hat{\mathbf{x}}, \hat{\boldsymbol{\theta}}, \mathcal{D}_{t-1})$. Therefore, we can simultaneously obtain maximally informative designs and optimal variational posteriors by jointly optimising the EIG lower bound w.r.t. the simulator inputs and the variational distribution as:

$$\hat{\mathbf{x}}_t, \hat{\boldsymbol{\theta}}_t, q_t \in \underset{\hat{\mathbf{x}} \in \mathcal{X}, \hat{\boldsymbol{\theta}} \in \Theta, q \in \mathcal{Q}}{\operatorname{argmax}} \widehat{\mathrm{EIG}}_t(\hat{\mathbf{x}}, \hat{\boldsymbol{\theta}}, q) = \underset{\hat{\mathbf{x}} \in \mathcal{X}, \hat{\boldsymbol{\theta}} \in \Theta, q \in \mathcal{Q}}{\operatorname{argmax}} \mathbb{E}_{p(\hat{y}, \boldsymbol{\theta}^*|\hat{\mathbf{x}}, \hat{\boldsymbol{\theta}}, \mathcal{D}_{t-1})}[\log q(\boldsymbol{\theta}^*|\hat{y})], \tag{17}$$

given a suitable variational family $\mathcal{Q}$ of conditional distributions. Note that, in this formulation, we only need samples from the posterior $p(\boldsymbol{\theta}^*|\mathcal{D}_{t-1})$ to estimate the expectation above, which can be approximated via Monte Carlo, without requiring densities other than that of the variational model $q$.

## 5.2 Algorithm

Algorithm 1 summarises the method we propose, which we name *Bayesian Adaptive Calibration and Optimal desigN* (BACON). The algorithm starts with an initial dataset $\mathcal{D}_0$ containing the real data (and possibly previously available simulation data) and an estimate of the posterior given the initial data $p(\boldsymbol{\theta}^*|\mathcal{D}_0)$. Posterior estimates in BACON can be represented by samples obtained via Markov chain Monte Carlo (MCMC) or variational inference over the GP model and the currently available data $\mathcal{D}_t$. Note that we only need samples from the previous posterior to estimate the expectation in Eq. 17, with no need to directly evaluate its probability densities. Each iteration starts by optimising the variational EIG lower bound using the objective in Eq. 17 to jointly select an optimal design $\hat{\mathbf{x}}_t$, simulation parameters $\hat{\boldsymbol{\theta}}_t$ and variational posterior $q_t$. Given the new design $\hat{\mathbf{x}}_t$, we run the simulation with the chosen parameters $\hat{\boldsymbol{\theta}}_t$, observing a new outcome $\hat{y}_t$. The calibration posterior $p_t(\boldsymbol{\theta}^*)$ and the GP model are then updated with the new data, potentially including a re-estimation of the GP hyper-parameters via, for example, maximum likelihood estimation. The process then repeats given the updated GP and posterior for up to a given number of iterations $T$. At the end, a final posterior $p_T(\boldsymbol{\theta}^*) = p(\boldsymbol{\theta}^*|\mathbf{y}_R, \hat{\mathbf{y}}_T)$ and a conditional density model $q_T$ are obtained.

---

[3]We will at times write $q(\boldsymbol{\theta}^*|\hat{y})$ to denote $q(\boldsymbol{\theta}^*|\hat{y}, \hat{\mathbf{x}}, \hat{\boldsymbol{\theta}})$ to avoid notation clutter, as the dependence on the inputs $(\hat{\mathbf{x}}, \hat{\boldsymbol{\theta}})$ remains implicit through the conditioning on $\hat{y}$.

**Algorithm 1** BACON

**input** $\mathcal{D}_0 := \{\mathcal{X}_R, \mathbf{y}_R\}$;  {Real data}
**input** $p_0(\boldsymbol{\theta}^*) := p(\boldsymbol{\theta}^*|\mathcal{D}_0)$  {MCMC or VI prior distribution}
  **for** $t \in \{1, \ldots, T\}$ **do**
    $\hat{\mathbf{x}}_t, \hat{\boldsymbol{\theta}}_t, q_t \in \operatorname{argmax}_{\hat{\mathbf{x}}, \hat{\boldsymbol{\theta}}, q} \mathbb{E}_{p_{t-1}(\hat{y}, \boldsymbol{\theta}^*|\hat{\mathbf{x}}, \hat{\boldsymbol{\theta}})}\left[\log q(\boldsymbol{\theta}^*|\hat{y})\right]$  {Optimise $\widehat{\mathrm{EIG}}_t$}
    $\hat{y}_t := h(\hat{\mathbf{x}}_t, \hat{\boldsymbol{\theta}}_t)$  {Run simulation}
    $\mathcal{D}_t := \mathcal{D}_{t-1} \cup \{\hat{\mathbf{x}}_t, \hat{\boldsymbol{\theta}}_t, \hat{y}_t\}$  {Update GP model}
    $p_t(\boldsymbol{\theta}^*) = p(\boldsymbol{\theta}^*|\mathcal{D}_{t-1})$  {Update posterior via MCMC or VI}
  **end for**
**output** $p_T(\boldsymbol{\theta}^*)$  {Final posterior}

## 5.3 Variational posteriors

Any conditional probability density model $q(\boldsymbol{\theta}^*|\hat{y})$ estimating probability densities over the parameter space $\Theta$ given an observation $\hat{y}$ could suit our method. In the following, we describe two possible parameterisations for this model. The first facilitates marginalising latent inputs in GP regression [31, 32], while the second better captures multi-modality in the posterior.

**Conditional Gaussian models.** Assuming we can approximate $p(\boldsymbol{\theta}^*|\mathcal{D}_t)$ as a Gaussian, we can construct a variational conditional density model as:

$$q_{\boldsymbol{\phi}}(\boldsymbol{\theta}^*|\hat{y}, \hat{\mathbf{x}}, \hat{\boldsymbol{\theta}}) := \mathcal{N}(\boldsymbol{\theta}^*; \mathbf{m}_{\boldsymbol{\phi}}(\hat{y}, \hat{\mathbf{x}}, \hat{\boldsymbol{\theta}}), \boldsymbol{\Sigma}_{\boldsymbol{\phi}}(\hat{y}, \hat{\mathbf{x}}, \hat{\boldsymbol{\theta}})), \tag{18}$$

where $\mathbf{m}_{\boldsymbol{\phi}}$ and $\boldsymbol{\Sigma}_{\boldsymbol{\phi}}$ are given by parametric models, such as neural networks, with parameters $\boldsymbol{\phi}$. To ensure $\boldsymbol{\Sigma}_{\boldsymbol{\phi}}(\cdot)$ is positive-definite, it can be parameterised by its Cholesky decomposition $\boldsymbol{\Sigma}_{\boldsymbol{\phi}}(\cdot) = \mathbf{L}_{\boldsymbol{\phi}}(\cdot)\mathbf{L}_{\boldsymbol{\phi}}(\cdot)^{\mathsf{T}}$, where $\mathbf{L}_{\boldsymbol{\phi}}(\cdot)$ is a lower-triangular matrix with positive diagonal entries.

**Conditional normalising flows** Normalising flows [33] apply the change-of-variable formula to derive composable, invertible transformations $\mathbf{g}_{\mathbf{w}}$ of a fixed base distribution $p_0$:

$$\mathbf{g}_{\mathbf{w}}(\boldsymbol{\xi}_0) := \mathbf{g}_{\mathbf{w}}^{(K)} \circ \cdots \circ \mathbf{g}_{\mathbf{w}}^{(1)}(\boldsymbol{\xi}_0), \quad \boldsymbol{\xi}_0 \sim p_0 \tag{19}$$

The log-probability density of a point $\boldsymbol{\xi} = \mathbf{g}_{\mathbf{w}}(\boldsymbol{\xi}_0)$ under this model can be calculated as:

$$\log p_K(\boldsymbol{\xi}; \mathbf{w}) = \log p_0(\boldsymbol{\xi}_0) - \sum_{j=1}^{K} \log \left|\mathbf{J}_{\mathbf{w}}^{(j)}(\boldsymbol{\xi}_{j-1})\right|,$$

where $\boldsymbol{\xi}_0 := \mathbf{g}_{\mathbf{w}}^{-1}(\boldsymbol{\xi})$, $\boldsymbol{\xi}_j := \mathbf{g}_{\mathbf{w}}^{(j)}(\boldsymbol{\xi}_{j-1})$, and $\mathbf{J}_{\mathbf{w}}^{(j)}$ is the Jacobian matrix of the $j$th transform $\mathbf{g}_{\mathbf{w}}^{(j)}$, for $j \in \{1, \ldots, K\}$. Several invertible flow architectures have been proposed in the literature, including radial and planar flows [33], autoregressive models [34–36] and models based on splines [37].

To derive a conditional density model $q_{\boldsymbol{\phi}}(\boldsymbol{\theta}^*|\hat{y})$, conditional normalising flows map the original flow parameters $\mathbf{w}$ via a neural network model $\mathbf{r}_{\boldsymbol{\phi}}: \hat{y} \mapsto \mathbf{w}$ [38, 39]. The resulting variational conditional density model is then given by:

$$\log q_{\boldsymbol{\phi}}(\boldsymbol{\theta}^*|\hat{y}, \hat{\mathbf{x}}, \hat{\boldsymbol{\theta}}) = \log p_K(\boldsymbol{\theta}^*; \mathbf{r}_{\boldsymbol{\phi}}(\hat{y}, \hat{\mathbf{x}}, \hat{\boldsymbol{\theta}})). \tag{20}$$

## 5.4 Batch parallel evaluations

Often simulations can be run in parallel by spawning multiple processes in a single machine or over a high-performance computing cluster. In this case, proposing batches with multiple simulation inputs can be more effective than running single simulations in a sequence. Optimising the EIG w.r.t. a batch of inputs $\mathcal{B} := \{\hat{\mathbf{x}}_i, \hat{\boldsymbol{\theta}}_i\}_{i=1}^{B}$, instead of single points, we obtain a batch version of Algorithm 1. In this case, we are seeking a batch that maximises the mutual information between the parameters $\boldsymbol{\theta}^*$ and the resulting simulation outcomes, i.e.:

$$\mathrm{EIG}_t(\mathcal{B}) = \mathbb{I}(\boldsymbol{\theta}^*; \{\hat{y}_i\}_{i=1}^{B}|\mathcal{B}, \mathcal{D}_{t-1}) \geq \mathbb{E}_{p(\{\hat{y}_i\}_{i=1}^{B}, \boldsymbol{\theta}^*|\mathcal{B}, \mathcal{D}_{t-1})}\left[\log \frac{q(\boldsymbol{\theta}^*|\{\hat{y}_i\}_{i=1}^{B})}{p(\boldsymbol{\theta}^*|\mathcal{D}_{t-1})}\right] \tag{21}$$

We optimise this objective with variational models that accept multiple conditioning observations $q(\boldsymbol{\theta}^*|\hat{y}_1, \ldots, \hat{y}_B)$. In practice, this simply amounts to replacing the single conditioning entries to the models in Sec. 5.3 by the concatenated batch or a permutation-invariant deep set encoding [16, 40].

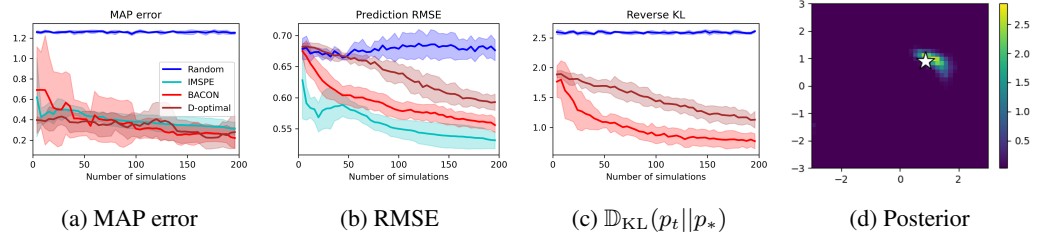

(a) MAP error      (b) RMSE      (c) $\mathbb{D}_{\mathrm{KL}}(p_t||p_*)$      (d) Posterior

Figure 1: Experimental results on synthetic data where the target posterior $p^*$ is unimodal. The first 3 plots show estimates for performance metrics as a function of the number of simulations run (not including the initial data). Estimates were computed based on the posterior estimates for each method available during their run, with *random* using $p(\boldsymbol{\theta}^*)$, D-optimal and BACON using MCMC posteriors, and IMSPE using a Dirac delta (reverse KL undefined, not shown) on the MAP estimate as posterior estimates. Results are averaged over 10 trials, and shaded areas indicate $\pm 1$ standard deviation. The rightmost plot shows the target posterior, with the true $\boldsymbol{\theta}^*$ indicated by a star.

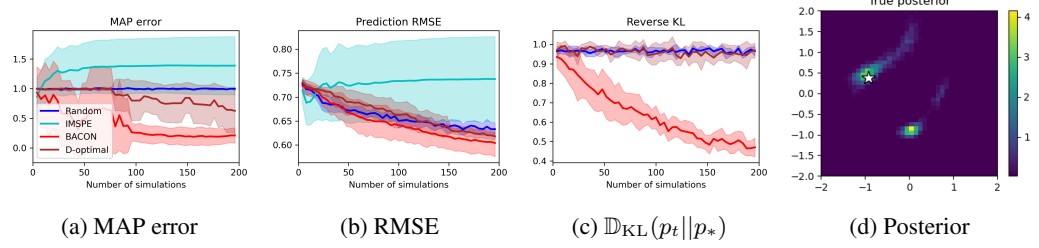

(a) MAP error      (b) RMSE      (c) $\mathbb{D}_{\mathrm{KL}}(p_t||p_*)$      (d) Posterior

Figure 2: Experimental results on synthetic data where the target posterior $p^*$ is bimodal. See Fig. 1 for details, with the exception that the rightmost plot now shows the bimodal target posterior.

## 6 Experiments

In this section, we present experimental results on synthetic and real-data problems evaluating the proposed variational Bayesian adaptive calibration framework against baselines. Further experimental details can be found in Appendix A and in our code repository.[4]

**Performance metrics.** We evaluated each method against a set of performance metrics, which we now describe. The maximum-a-posteriori (MAP) error measures the distance between the mode of the variational distribution and the true parameters $\boldsymbol{\theta}^*$. To measure the quality of the learnt model in predicting real outcomes, we also evaluated the root mean square error (RMSE) between the expected GP predictions under the learnt variational distribution and real outcomes: $\mathrm{RMSE} := \sqrt{\frac{1}{N}\sum_{i=1}^{N}(\mathbb{E}_{q(\boldsymbol{\theta})}[\mu(\mathbf{x}_i^*, \boldsymbol{\theta}^*; \boldsymbol{\theta})] - y_i^*)^2}$, where $y_i^* = f(\mathbf{x}_i^*) + \nu_i^*$ are observations of the real process over a set of test points $\{\mathbf{x}_i^*\}_{i=1}^{N} \subset \mathcal{X}$ placed on a uniform grid over the design space.

**Information gain.** Lastly, we also evaluated two sample-based estimates of the KL divergence [41]. Namely, $\mathbb{D}_{\mathrm{KL}}(p_T||p_0)$ corresponds to the KL divergence between the final MCMC posterior (given all simulations and real data) and the initial one (given only the real data and an initial set of randomised simulations) both estimated over the learnt GP model. The column $\mathbb{D}_{\mathrm{KL}}(p_T||p^*)$ indicates the KL divergence between the final MCMC posterior $p_T$ and the posterior $p^*$ with full knowledge of the simulator, which can be cheaply evaluated in this synthetic scenario. The average of $\mathbb{D}_{\mathrm{KL}}(p_T||p_0)$ is an indicator for the expected information gain (1) of an algorithm, given that it is the expected relative entropy across the possible trajectories of observations. In contrast, $\mathbb{D}_{\mathrm{KL}}(p_T||p^*)$ indicates how far the estimates are from the best possible posterior obtainable with a model that is given the available real data and (a potentially infinite amount of) simulations.

---

[4]Code available at: `https://github.com/csiro-funml/bacon`

| | $\mathbb{D}_{\mathrm{KL}}(p_T\|p_0)$ ↑ | $\mathbb{D}_{\mathrm{KL}}(p_T\|p^*)$ ↓ | | $\mathbb{D}_{\mathrm{KL}}(p_T\|p_0)$ ↑ | $\mathbb{D}_{\mathrm{KL}}(p_T\|p^*)$ ↓ |
|---|---|---|---|---|---|
| BACON | **1.00 ± 0.06** | 0.76 ± 0.13 | BACON | **0.40 ± 0.03** | **0.45 ± 0.06** |
| IMSPE | 0.89 ± 0.11 | 1.05 ± 0.19 | IMSPE | 0.19 ± 0.04 | 0.70 ± 0.07 |
| D-optim. | 0.42 ± 0.11 | 1.09 ± 0.15 | D-optim. | 0.07 ± 0.02 | 0.94 ± 0.03 |
| Random | 0.62 ± 0.07 | 1.18 ± 0.13 | Random | 0.28 ± 0.07 | 0.54 ± 0.07 |
| VBMC | – | **0.53 ± 0.02** | VBMC | – | 0.49 ± 0.13 |

(a) Unimodal posterior           (b) Bimodal posterior

Table 1: Results for 2+2D synthetic problem after $T = 50$ iterations (batch of $B = 4$). Here $\mathbb{D}_{\mathrm{KL}}(p_T\|p_0)$ corresponds to the KL divergence between the final posterior (estimated after each algorithm's run with all the data it collected) and the starting one (higher is better), while $\mathbb{D}_{\mathrm{KL}}(p_T\|p^*)$ is the KL between the final posterior and the posterior with full knowledge of the simulator $p^*$ (lower is better). All posteriors were sampled via MCMC using 4000 samples. Averages and standard deviations were estimated from 10 independent runs.

## 6.1 Baselines

Our algorithmic baselines were chosen to illustrate the main approaches currently available in the literature. All baselines are implemented as sequential methods, in the sense that their GP models are updated with the latest batch of observations before proceeding to the next iteration.

**Random search.** This baseline samples simulation designs $\hat{\mathbf{x}}_t \sim \mathcal{U}(\mathcal{X})$ from a uniform distribution over the design space $\mathcal{X}$ and calibration parameters from the prior $\hat{\boldsymbol{\theta}}_t \sim p(\boldsymbol{\theta}^*)$.

**IMSPE with MAP estimates.** The integrated mean squared prediction error (IMSPE) [42] criterion chooses designs $\hat{\mathbf{x}}_t$ and calibration $\hat{\boldsymbol{\theta}}_t$ parameters by minimising the GP prediction error:

$$\mathrm{IMSPE}_t(\hat{\mathbf{z}}) := \int_{\mathcal{Z}} \mathbb{E}[(\hat{f}(\mathbf{z}) - \mu_{t+1}(\mathbf{z};\boldsymbol{\theta}^*))^2 \mid \hat{f}(\hat{\mathbf{z}}), \mathcal{D}_t]\, \mathrm{d}\mathbf{z} = \int_{\mathcal{Z}} \sigma_{t+1}^2(\mathbf{z};\boldsymbol{\theta}^*|\mathcal{D}_t, \hat{f}(\hat{\mathbf{z}}))\, \mathrm{d}\mathbf{z}. \quad (22)$$

The posterior MAP estimate $\boldsymbol{\theta}_t^* \in \mathrm{argmax}_{\boldsymbol{\theta}}\, p(\boldsymbol{\theta}|\mathcal{D}_{t-1})$ is used as a point estimate for the true $\boldsymbol{\theta}^*$. The integral is approximated as a sum over a uniform grid of designs and samples from the calibration prior,[5] making IMSPE equivalent to active learning Cohn [43] and also a form of A-optimality [28].

**D-optimal designs.** We provide experimental results with an additional baseline following a D-optimality criterion, a classic experimental design objective. Optimal candidate designs according to this criterion are points of maximum uncertainty according to the model [28]. If we model the simulator as the unknown variable of interest, this corresponds to selecting designs where we have maximum entropy of the Gaussian predictive distribution $p(\hat{y}|\hat{\mathbf{x}}, \hat{\boldsymbol{\theta}}, \mathcal{D}_{t-1})$. This approach, therefore, simply attempts to collect an informative set of simulations according to the GP prior over the simulator $h$ only, without considering the information in the real data. Running D-optimality on $\boldsymbol{\theta}^*$, instead, would lead back to the EIG criterion we use.

**Variational Bayesian Monte Carlo (VBMC).** Acerbi [44] presents an adaptive Bayesian quadrature method to learn posterior distributions over models with black-box likelihood functions. The method estimates the posterior $p(\boldsymbol{\theta}^*|\mathbf{y}_R, h)$ by modelling the log-joint $\log p(\mathbf{y}_R, \boldsymbol{\theta}^*|h)$ as a sample from a Gaussian process. VBMC then learns a variational posterior approximation by maximising a lower-confidence bound over the ELBO given by the GP estimates. Calibration parameter queries $\hat{\boldsymbol{\theta}}_t$ are obtained by optimising quadrature-based acquisition functions. Regarding design points, simulations are always run on the set of real design points $\mathcal{X}_R$ in the observed data, which is fixed.

## 6.2 Synthetic experiments

For this experiment, we sampled a function $\hat{f} \sim \mathcal{GP}(0, k)$ to use as our simulator and compared different algorithms. Following a sparse GP approach [45], a function sampled from a GP can

---

[5]The original paper proposed analytic solutions to Eq. 22 tailored for specific kernels. However, we decided to keep our codebase generic to work with different kernels, and therefore opted for a numerical approximation.

|  | $\mathbb{D}_{\mathrm{KL}}(p_T\|p_0)\uparrow$ | $\mathbb{D}_{\mathrm{KL}}(p_T\|p^*)\downarrow$ |
|---|---|---|
| BACON | **0.37 ± 0.09** | **0.07 ± 0.06** |
| IMSPE | 0.22 ± 0.11 | 0.45 ± 0.21 |
| D-optimal | 0.21 ± 0.08 | 0.23 ± 0.10 |
| Random | 0.32 ± 0.09 | 0.20 ± 0.14 |
| VBMC | – | 5.48 ± 1.66 |

Table 2: Results on the location finding problem after $T = 30$ iterations with $B = 4$, $R = 20$ "real" data points and an initial set of 20 simulations. Estimates were averaged over 10 independent runs.

be approximated as $\hat{f}(\mathbf{z}) \approx k(\mathbf{z}, \mathbf{Z}_M)\mathbf{K}_M^{-1}\mathbf{u}_M$, where $\mathbf{u}_M \sim \mathcal{N}(\hat{\mathbf{u}}_M, \boldsymbol{\Sigma}_M)$ is a sample from an $M$-dimensional Gaussian, $\mathbf{Z}_M := \{\mathbf{z}_i\}_{i=1}^M \subset \mathcal{X} \times \Theta \times \{0, 1\}$, for a given $M$. As the number of points $M \to \infty$, if the pseudo-inputs $\mathbf{Z}_M$ form a dense set, the approximate $\hat{f}$ should converge in distribution to a sample from the Gaussian process $\mathcal{GP}(0, k)$. In our case, to sample $\mathbf{Z}_M$, we sample designs from a uniform distribution over the design space, calibration parameters from the prior, and fidelities from a Bernoulli distribution with parameter set to $0.5$. We also set $\hat{\mathbf{u}}_M := \mathbf{0}$ and $\boldsymbol{\Sigma}_M := \mathbf{K}_M = k(\mathbf{Z}_M, \mathbf{Z}_M)$. We repeatedly run a loop of $T$ iterations for each algorithm, with different random seeds.

We run each algorithm for $T := 50$ iterations using a batch of $B := 4$ designs per iteration. Each of the methods using GP approximations for the simulator are initialised with 20 observations and $R = 5$ real data points. To configure VBMC, we allow it to run an equivalent maximum amount of objective function evaluations. The design space is set as the 2-dimensional unit box $\mathcal{X} := [0, 1]^2$ and the "true" parameters are sampled from a standard normal prior $p(\boldsymbol{\theta}^*) := \mathcal{N}(\boldsymbol{\theta}^*; \mathbf{0}, \mathbf{I})$ also over a 2D space, totalling a 4-dimensional problem space.

Results are presented in Fig. 1 and 2. Fig. 1 shows a case where the GP-sampled simulator led to a unimodal target posterior. In this case, we see that BACON is able to achieve fast convergence in terms of MAP estimates and KL divergence towards the target posterior, while IMSPE dominates in terms of simulator approximation error as measured by the RMSE. As the posterior is unimodal and quite concentrated around the true parameter, it is natural that a method relying on MAP estimates, such as RMSE, would perform well. In contrast, when the posterior is multimodal, as shown in the bimodal case in Fig. 2, MAP estimates are not necessarily reliable any more, as they might get stuck on a non-informative mode, leading to biased estimates for IMSPE and a significant drop in performance. Lastly, note that D-optimal and random designs can also lead to RMSE approaching the lowest (as determined by the noise level with $\sigma_\nu = 0.5$) in some circumstances. However, these approaches do not directly provide posterior approximations and may fail in more complex scenarios.

In terms of final posterior estimates, Table 1 shows that VBMC estimates reach the closest to the full-knowledge target posterior $p^*$ in the unimodal case, while BACON is able to surpass the other GP-emulation approaches in terms of information gain. For the bimodal case, however, we see that BACON gains an advantage over VBMC. Recall that VBMC relies on a variational mixture of Gaussian distributions, while BACON applies conditional normalising flows for its posterior approximations, which lead to increased flexibility. In addition, despite the slightly worse performance than VBMC, BACON also provides a GP model that can be used as an emulator for the simulator (and to approximate the real process), while VBMC's focus is on approximating the log-likelihood.

### 6.3 Finding the location of hidden sources

We consider the problem of finding the location of 2 hidden sources in a 2D environment following the setting in Foster et al. [16]. We are provided with $R = 20$ initial measurements and an initial set of $S = 20$ randomised simulations without knowledge of the true parameters which the data was generated with. Sources are sampled from a standard normal, the design space is limited to the unit box, and noise is sampled with $\sigma_\nu = 0.5$. Our results are presented in Table 2, showing a similar tendency in higher information gain for our method, and a very low KL w.r.t. $p^*$. Note that a higher information gain indicates a more informative posterior, whose entropy will be much lower relative to the starting distribution, compared to the other methods. In addition, the ideal $p^*$, which a GP-based posterior should converge to in the limit of infinite data, is not known by the methods, only $p_0$. Therefore, besides obtaining maximally informative data, we have shown that BACON is also efficient in approximating posteriors over black-box simulators, while also learning a GP emulator.

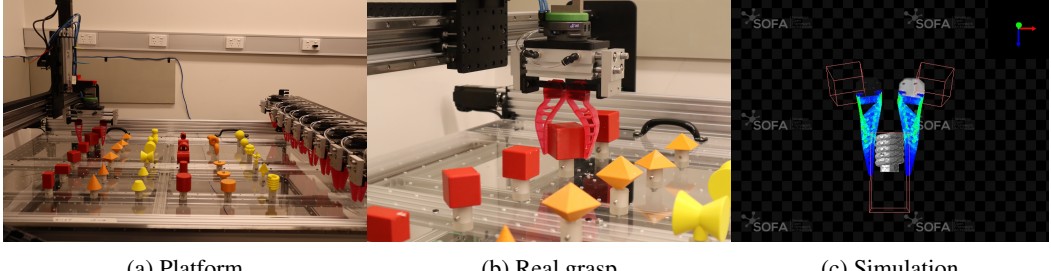

| | (a) Platform | (b) Real grasp | (c) Simulation |
|---|---|---|---|

Figure 3: Soft-robotics grasping experiment. We calibrate a soft materials simulator against real data from physical grasping from an automated experimentation platform

| | $\mathbb{D}_{\mathrm{KL}}(p_T\|p^*)\downarrow$ |
|---|---|
| BACON | $\mathbf{1.32\pm0.05}$ |
| IMSPE | $1.56\pm0.08$ |
| D-optimal | $1.50\pm0.05$ |
| Random | $1.48\pm0.07$ |

Table 3: Soft-robotics simulator calibration final results after $T=10$ with $B=16$ points per batch. The target posterior $p^*$ was inferred using a large set of 1024 random simulations uniformly covering the design and parameter space. Performance was averaged over 4 independent runs.

### 6.4 Soft-robotic grasping simulator calibration

For this experiment, we are provided with a dataset containing $R=10$ real measurements of the peak grasping force of soft robotic gripper designs on a range of testing objects (see Fig. 3). The gripper designs follow a fin-ray pattern parameterised by 9 geometric parameters [46], and we are interested in estimating 2 unknown physics parameters, the Young's modulus of elasticity and the coefficient of static friction with the objects. To simulate the gripper designs, we use the SOFA framework [47] to reproduce the grasping scenario and provide an estimate of the peak grasping force. In particular, for this paper, we focus on the grasping of a spherical object, which provides a simpler geometry and lower discrepancy with respect to real data measurements compared to more complex objects. This experiment provides us with a benchmark where simulations are expensive to run, taking from minutes to a few hours to run (depending on mesh resolution) on a high-performance computing platform. Therefore, it is important to choose a minimum amount of informative simulations.

Our results are shown in Table 3. Each algorithm was initialised with a set of 123 random simulations and run for $T=10$ iterations. The results show that BACON achieves the closest approximation to the target posterior. IMSPE highly concentrated its parameter choices around its posterior mode estimate, while other baselines were too spread, both leading to inferior posterior approximations (see Fig. 4 in the appendix) and showing the advantage of BACON's joint optimisation and inference.

## 7 Conclusion, limitations and future work

We have developed BACON, a Bayesian approach that carries out parameter calibration of computer models and optimal design of experiments *jointly*. It does so by optimizing an information-theoretic criterion so that input designs and calibration parameters are selected to be maximally informative about the optimal parameters. Our method provides a full posterior over optimal calibration parameters as well as an accurate Gaussian process based estimation of the computer model (i.e., an emulator). One of the main limitations of the presented framework, however, is scalability to large datasets, due to the cubic computational complexity of exact inference with GPs. A potential extension with scalable sparse variational GP models [48] using a conditional distribution model for the inducing points is discussed in Sec. B.2. We emphasize that our proposed method is still applicable to many real practical settings, where the problem constraints do not demand a very large number of simulation samples. Lastly, we also note that the method can be adapted to work with vector-valued observations by the use of multi-output GP models [49]. Further discussions on limitations and future work can be found in our appendix (see Appendix B and C).

## Acknowledgements

This project was supported by resources and expertise provided by CSIRO IMT Scientific Computing. We are also grateful for the support of CSIRO's Data61 soft-robotics team, especially Josh Pinskier, Xing Wang, Lois Liow, Sarah Baldwin, James Brett and Vinoth Viswanathan, in the experimental data collection and simulations setup for the soft-robotics calibration problem.

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

**Algorithm 2** BACON (split training)

---

$\mathcal{D}_0 := \{\mathcal{X}_R, \mathbf{y}_R\};$
**for** $t \in \{1, \dots, T\}$ **do**
  $\mu_{t-1}, k_{t-1} \leftarrow \text{UpdateGP}(\mathcal{D}_{t-1})$
  $\{\boldsymbol{\theta}_i^*\}_{i=1}^{S_A} \overset{\text{MCMC}}{\sim} p(\boldsymbol{\theta}^*|\mathcal{D}_{t-1}) \propto p(\boldsymbol{\theta}^*)\mathcal{N}(\mathbf{y}_R; \boldsymbol{\mu}_{t-1}(\mathbf{Z}_R(\boldsymbol{\theta}^*); \boldsymbol{\theta}^*), \boldsymbol{\Sigma}_{t-1}(\mathbf{Z}_R(\boldsymbol{\theta}^*); \boldsymbol{\theta}^*) + \sigma_\nu^2 \mathbf{I})$
  $p(\boldsymbol{\theta}^*|\mathcal{D}_{t-1}) \approx \hat{p}_{t-1} := \frac{1}{S_A}\sum_{i=1}^{S_A} \delta_{\boldsymbol{\theta}_i^*}$
  $q_t \leftarrow \text{TrainFlow}(\hat{p}_{t-1}, \mathcal{D}_{t-1})$
  $\{\hat{\mathbf{x}}_{t,i}, \hat{\boldsymbol{\theta}}_{t,i}\}_{i=1}^{B} \leftarrow \text{OptimiseDesigns}(q_t, \hat{p}_{t-1}, \mathcal{D}_{t-1})$
  $\hat{y}_{t,i} := h(\hat{\mathbf{x}}_{t,i}, \hat{\boldsymbol{\theta}}_{t,i})$ (parallel) for $i \in \{1, \dots, B\}$          {Run batch of simulations}
  $\mathcal{D}_t := \mathcal{D}_{t-1} \cup \{\hat{\mathbf{x}}_{t,i}, \hat{\boldsymbol{\theta}}_{t,i}, \hat{y}_{t,i}\}_{i=1}^{B}$          {Update GP dataset}
**end for**

---

# A    Additional details on the experiments

For all experiments, we use conditional normalising flows as the variational model for BACON. Our implementation for BACON and most of the baselines, except for VBMC,[6] is based on Pyro probabilistic programming models [50]. Gaussian process modelling code is based on BoTorch[7] [51]. The flow architecture is chosen for each synthetic-data problem by running hyper-parameter tuning with a simplified version of the problem. Most Gaussian process models are parameterised with Matérn kernels [2, Ch. 4] and constant or zero mean functions. Pyro's MCMC with its default no-U-turn (NUTS) sampler [52] was applied to obtain samples from $p(\boldsymbol{\theta}^*|\mathcal{D}_{t-1})$ at each iteration $t$. KL divergences are computed from samples using a nearest neighbours estimator implemented in the information theoretical estimators (ITE) package[8] [41].

## A.1    Synthetic GP problem

The GP prior was set with $\hat{k}$ given by a squared exponential kernel and $k_\varepsilon$ given by a Matérn kernel with smoothness parameter set to $2.5$ [2]. The conditional normalising flow was configured with 2 layers of neural spline flows [37]. Batches of arbitrary size are used for conditioning via a permutation invariant set encoder, similar to Blau et al. [17], with a 2-layer, 32-units-wide fully-connected hyperbolic tangent neural network passing through a summation at the end. Gradient-based optimisation is run using Adam with a learning rate $10^{-3}$ for the flow parameters and $0.05$ for the simulation design points, both using cosine annealing with warm restarts as a learning rate scheduler. 256 samples were subsampled from the MCMC posterior to estimate expectations for both this and the location-finding problem.

**Algorithm with split training.**    For the synthetic GP problem, we provide a more detailed pseudo-code of our algorithmic implementation using an option for training the conditional normalising flow and optimising the designs separately. Specifically, we applied MCMC to estimate our posteriors and had a flexible optimisation loop, where we had the option to separate the training of the conditional normalising flow model from the optimisation of the design points, as shown in Algorithm 2. This approach can make the algorithm more stable, though at the cost of a longer runtime. This option was only applied to the GP-based synthetic experiments, while for the other experiments we ran the full joint optimisation over both the simulation inputs $(\hat{\mathbf{x}}, \hat{\boldsymbol{\theta}})$ and the variational parameters of the conditional model $q$.

## A.2    Location finding problem

For this experiment we used more up-to-date Zuko[9] implementations of the conditional normalising flow models, which were again set as neural spline flows [37] combined with a set encoder to condition on arbitrary batch sizes. Further architectural details can be found in our code repository. 256 samples

---

[6]For VBMC, we used its author's Python implementation at: `https://github.com/acerbilab/pyvbmc`
[7]BoTorch: `https://botorch.org`
[8]ITE package: `https://bitbucket.org/szzoli/ite-in-python`
[9]Zuko: `https://zuko.readthedocs.io/stable/`

---

**Algorithm 3** TrainFlow

---

**input** $\hat{p}_t, \mathcal{D}_t$
   **for** $n \in \{1, \ldots, N\}$ **do**
      $\{\hat{\boldsymbol{\theta}}_i\}_{i=1}^B \sim (1 - \epsilon)\hat{p}_t + \epsilon p$
      $\{\hat{\mathbf{x}}_i\}_{i=1}^B \sim \mathcal{U}(\mathcal{X})$
      $\{\boldsymbol{\theta}_i^*\}_{i=1}^S \sim \hat{p}_t$
      $\{\hat{y}_{i,j}\}_{i,j=1}^{S,B} \sim \mathcal{N}(\boldsymbol{\mu}_t(\{\hat{\mathbf{x}}_i, \hat{\boldsymbol{\theta}}_i\}_{i=1}^B; \{\boldsymbol{\theta}_i^*\}_{i=1}^S), \boldsymbol{\Sigma}_t(\{\hat{\mathbf{x}}_i, \hat{\boldsymbol{\theta}}_i\}_{i=1}^B; \{\boldsymbol{\theta}_i^*\}_{i=1}^S))$
      $\phi \leftarrow \phi + \frac{\eta}{S} \sum_{i=1}^S \nabla_\phi \log q_\phi \left( \boldsymbol{\theta}_i^* \mid \mathcal{D}_t \cup \{\hat{\mathbf{x}}_j, \hat{\boldsymbol{\theta}}_j, \hat{y}_{i,j}\}_{j=1}^B \right)$
   **end for**
**output** $q_\phi$

---

---

**Algorithm 4** OptimiseDesigns

---

**input** $q_t, \hat{p}_{t-1}, \mathcal{D}_{t-1}$
   $\widehat{\boldsymbol{\Theta}} = \{\hat{\boldsymbol{\theta}}_i\}_{i=1}^B \sim (1 - \epsilon)\hat{p}_{t-1} + \epsilon p$
   $\hat{\mathbf{X}} = \{\hat{\mathbf{x}}_i\}_{i=1}^B \sim \mathcal{U}(\mathcal{X})$
   **for** $n \in \{1, \ldots, N\}$ **do**
      $\{\boldsymbol{\theta}_i^*\}_{i=1}^S \sim \hat{p}_{t-1}$
      $\{\hat{\mathbf{y}}_i\}_{i=1}^S \sim \mathcal{N}(\boldsymbol{\mu}_{t-1}(\hat{\mathbf{X}}, \widehat{\boldsymbol{\Theta}}; \{\boldsymbol{\theta}_i^*\}_{i=1}^S), \boldsymbol{\Sigma}_{t-1}(\hat{\mathbf{X}}, \widehat{\boldsymbol{\Theta}}; \{\boldsymbol{\theta}_i^*\}_{i=1}^S))$
      $(\hat{\mathbf{X}}, \widehat{\boldsymbol{\Theta}}) \leftarrow (\hat{\mathbf{X}}, \widehat{\boldsymbol{\Theta}}) + \frac{\eta}{S} \sum_{i=1}^S \nabla_{\hat{\mathbf{X}}, \widehat{\boldsymbol{\Theta}}} \log q_t \left( \boldsymbol{\theta}_i^* \mid \mathcal{D}_t \cup \{\hat{\mathbf{X}}, \widehat{\boldsymbol{\Theta}}, \hat{\mathbf{y}}_i\} \right)$
   **end for**
**output** $\{\hat{\mathbf{x}}_i, \hat{\boldsymbol{\theta}}_i\}_{i=1}^B$

---

were subsampled from the MCMC posterior at each iteration to estimate expectations for EIG lower bound computations. The simulations kernel $\hat{k}$ was a Matérn 2.5 kernel. For this experiment we did not model the error term, leaving it with a zero kernel, since data is generated directly from the simulator with no further error component, only Gaussian noise with a standard deviation of 0.5. Final KL estimates were computed using the maximum-a-posteriori hyper-parameters of the GP model learnt with the random search approach to minimise biases in the estimate of $\mathbb{D}_{\mathrm{KL}}(p_T || p_0)$ due to differing GP hyper-parameters across baselines.

### A.3 Soft-robotics simulation problem

The prior for the calibration parameters $p(\boldsymbol{\theta}^*)$ in this experiment consisted of a 2-dimensional standard normal transformed through a sigmoid and an affine transform composition to provide a smooth uniform distribution over a pre-specified range for the calibration parameters. Such smooth approximation allows gradients to be computed near the edges of the parameter space while not allowing optimisation to take the calibration parameter candidates outside the uniform prior boundaries, since these would be placed at infinity under the normalised space. The conditional normalising flow model used Zuko's implementation of neural spline flows with 10 transform layers. The set encoder consisted of a 2-layer fully connected 32-unit-wide neural network encoding each input into an 8-dimensional output which was then summed and passed through as the context input to condition the flow. Adam again was used for optimisation with a learning rate of 0.001 for the flow and 0.05 for the simulation inputs. Monte Carlo expectation estimates used 256 samples from the current MCMC posterior at each joint optimisation step.

### A.4 Hyper-parameter tuning

Besides the GP hyperparameters (e.g., lengthscales, noise variance, etc.), which had to be tuned for the non-GP-based problems, there are optimisation settings (i.e., step sizes, scheduling rates, etc.), conditional density model hyper-parameters (i.e., normalising flow architecture), and other algorithmic settings, e.g., the designs batch size $B$. The latter is dependent on the available computing resources (e.g., number of CPU cores or compute nodes for simulations in a high-performance computing system). We tuned optimisation settings and architectural parameters for the conditional

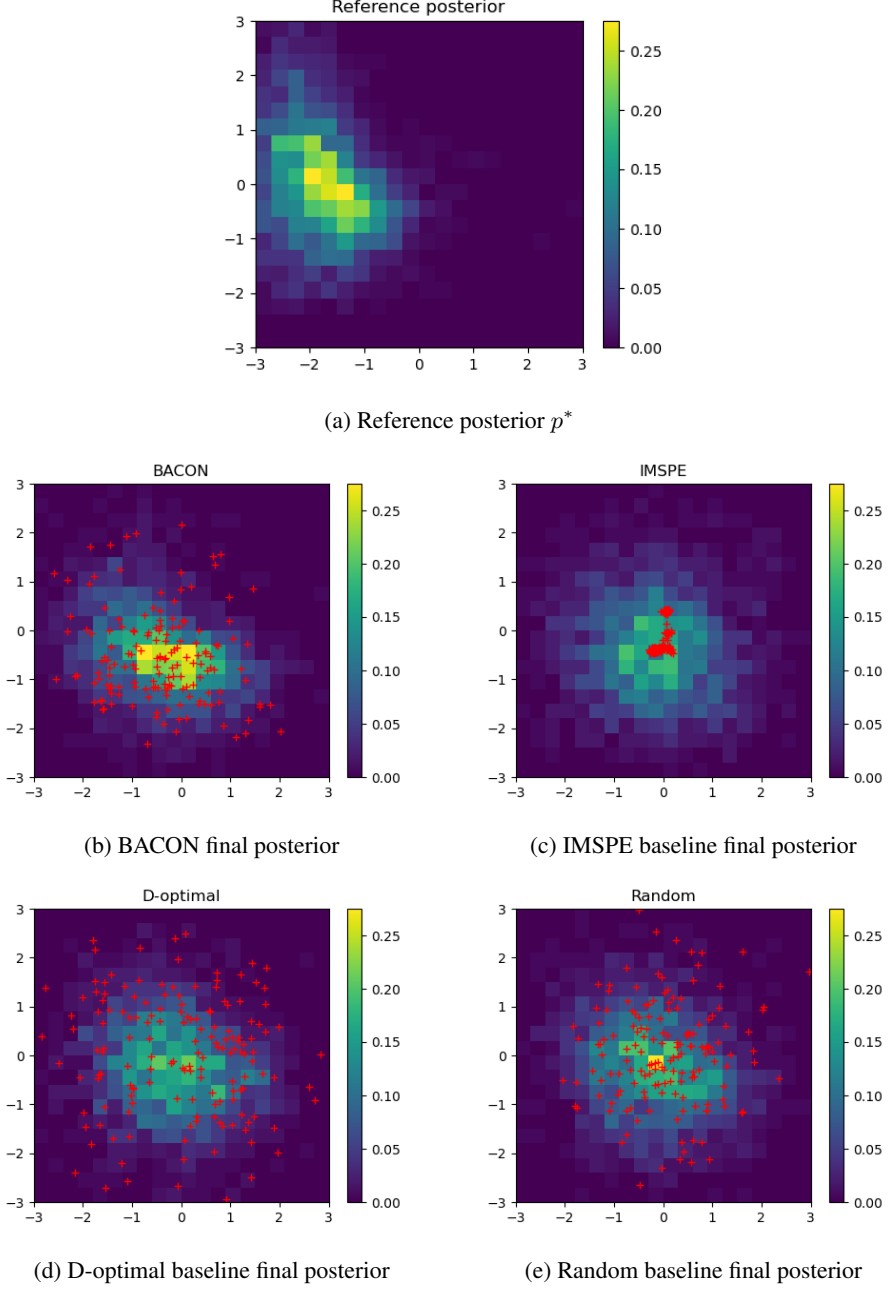

Figure 4: Final posterior approximations $p(\boldsymbol{\theta}^*|\mathcal{D}_T)$ and simulation parameter $\hat{\boldsymbol{\theta}}$ (red crosses) choices by each method for the soft-robotics simulator calibration problem after one of the runs. The target/reference posterior (a) was inferred using a large number (1024) of simulations following a Latin hypercube pattern over the combined design $\mathcal{X}$ and calibration parameters space $\Theta$ and a uniform prior $p(\boldsymbol{\theta})$ over the same range as the smooth uniform prior the algorithms used. The posteriors are plotted as a 2D histogram over the normalised range (after an affine and sigmoid transform), which the algorithms used for optimisation. The KL divergences in Table 3 are computed with respect to this reference posterior. Also note that the simulation parameters $\hat{\boldsymbol{\theta}}$ in the plot correspond to different algorithmic choices for design inputs $\hat{\mathbf{x}}$, which are 9-dimensional variables that are not plotted here.

normalising flows via Bayesian optimisation with short runs (e.g., 10-20 iterations) on the synthetic problem. However, depending on the number of parameters, a simpler approach, like grid search, might be enough. GP hyper-parameters were optimised online via maximum a posteriori estimation after each iteration's batch update. Further implementation details can be found in our code repository.[10]

## B  Extensions of the proposed approach

In the following, we present two extensions to deal with limitations of the current approach. Namely, we can amortise inference over the calibration posterior by reutilising the learnt conditional distribution models as priors, instead of having to run, for example, MCMC. Secondly, we present derivations for a scalable sparse GP version of our method.

### B.1  Amortisation

We use a conditional variational distribution model for $q(\boldsymbol{\theta}^*|\hat{y})$. The main advantage of training a conditional model is that, once new data $\hat{y}_t$ is observed, we readily obtain an approximation to the new posterior as $p(\boldsymbol{\theta}^*|\mathcal{D}_t) = p(\boldsymbol{\theta}^*|\hat{y}_t, \hat{\mathbf{x}}_t, \hat{\boldsymbol{\theta}}_t, \mathcal{D}_{t-1}) \approx q_t(\boldsymbol{\theta}^*|\hat{y}_t)$. There is, therefore, potential to reuse the variational posterior as the prior for the next iteration, and all the optimisation is concentrated within a single loop.

**Approximate objective.**  We are still left with terms dependent on the posterior from the previous iteration $p(\boldsymbol{\theta}^*|\mathcal{D}_{t-1})$ in Eq. 15. Firstly, however, note that the denominator inside the expectation is constant w.r.t. the optimisation variables, not affecting the maximiser. Secondly, we may replace the joint predictive distribution $p(\hat{y}, \boldsymbol{\theta}^*|\hat{\mathbf{x}}, \hat{\boldsymbol{\theta}}, \mathcal{D}_{t-1})$ by an approximation using the previous optimal variational posterior $q_{t-1}$ as:

$$p(\hat{y}, \boldsymbol{\theta}^*|\hat{\mathbf{x}}, \hat{\boldsymbol{\theta}}, \mathcal{D}_{t-1}) \approx q_{t-1}(\hat{y}, \boldsymbol{\theta}^*|\hat{\mathbf{x}}, \hat{\boldsymbol{\theta}}) := p(\hat{y}|\boldsymbol{\theta}^*, \hat{\mathbf{x}}, \hat{\boldsymbol{\theta}}, \mathcal{D}_{t-1})q_{t-1}(\boldsymbol{\theta}^*) \tag{23}$$

where $q_{t-1}(\boldsymbol{\theta}^*) := q_{t-1}(\boldsymbol{\theta}^*|\hat{y}_{t-1}) \approx p(\boldsymbol{\theta}^*|\mathcal{D}_{t-1})$. The following objective then approximately shares the same set of maximisers as the variational lower bound $\widehat{\mathrm{EIG}}_t(\hat{\mathbf{x}}, \hat{\boldsymbol{\theta}}, q)$:

$$\hat{\mathbf{x}}_t, \hat{\boldsymbol{\theta}}_t, q_t \in \underset{\hat{\mathbf{x}} \in \mathcal{X}, \hat{\boldsymbol{\theta}} \in \Theta, q \in \mathcal{Q}}{\operatorname{argmax}} \mathbb{E}_{q_{t-1}(\hat{y}, \boldsymbol{\theta}^*|\hat{\mathbf{x}}, \hat{\boldsymbol{\theta}})} \left[ \log q(\boldsymbol{\theta}^*|\hat{y}) \right] . \tag{24}$$

In practice, reusing the variational conditional posterior may tend to degenerate the approximation over time. However, that can be corrected by rerunning MCMC or a variational inference scheme over the data to obtain a fresh new posterior at every few iterations.

### B.2  Conditional sparse models for large datasets

Computing the variational EIG requires evaluating expectations with respect to the posterior predictive distribution $p(\hat{y}|\boldsymbol{\theta}^*, \hat{\mathbf{x}}, \hat{\boldsymbol{\theta}}, \mathcal{D}_t)$. Note, however, that, as $\boldsymbol{\theta}^*$ appears inside a matrix inversion in the GP predictive (Eq. 8), each sample of $p(\hat{y}|\boldsymbol{\theta}^*, \hat{\mathbf{x}}, \hat{\boldsymbol{\theta}}, \mathcal{D}_t)$ requires a $\mathcal{O}(N_t^3)$ computation cost, where $N_t := R + t$ is the number of data points at iteration $t \in \mathbb{N}$. This cost may quickly become prohibitive for reasonably large datasets, which are easily obtainable in batch settings (Sec. 5.4), rendering EIG computations infeasible. To scale our method to handle large amounts of data, we then need GP models that can reduce this computational complexity, while still allowing us to obtain reasonable EIG estimates.

### B.2.1  Variational sparse GP approximation

We consider an augmentation to the original GP model which allows us to sparsify its covariance matrix, reducing the computational complexity of GP predictions. Following the variational sparse GP approach [48], let $\mathbf{u} := \hat{f}(\mathbf{Z}_u) \in \mathbb{R}^M$ denote a vector of $M$ inducing variables representing unknown function values at a given set of pseudo-inputs $\mathbf{Z}_u$. The joint distribution between observations $\mathbf{y}$,

---

[10]Code available at: https://github.com/csiro-funml/bacon

function values $\hat{\mathbf{f}} := \hat{f}(\mathbf{Z}(\boldsymbol{\theta}^*))$, inducing variables $\mathbf{u}$ and the unknown parameters $\boldsymbol{\theta}^*$ can be written as:

$$p(\mathbf{y}, \hat{\mathbf{f}}, \mathbf{u}, \boldsymbol{\theta}^*) = p(\mathbf{y}, \hat{\mathbf{f}}, \mathbf{u}|\boldsymbol{\theta}^*)p(\boldsymbol{\theta}^*) = p(\mathbf{y}|\hat{\mathbf{f}})p(\hat{\mathbf{f}}|\mathbf{u}, \boldsymbol{\theta}^*)p(\mathbf{u})p(\boldsymbol{\theta}^*), \tag{25}$$

where $p(\mathbf{y}|\hat{\mathbf{f}}) = \mathcal{N}(\mathbf{y}; \hat{\mathbf{f}}, \boldsymbol{\Sigma}_{\mathbf{y}})$,

$$p(\hat{\mathbf{f}}|\mathbf{u}, \boldsymbol{\theta}^*) = \mathcal{N}(\hat{\mathbf{f}}; \mathbf{K}_{\hat{f}u}(\boldsymbol{\theta}^*)\mathbf{K}_{uu}^{-1}\mathbf{u}, \mathbf{K}_{\hat{f}\hat{f}}(\boldsymbol{\theta}^*) - \mathbf{K}_{\hat{f}u}(\boldsymbol{\theta}^*)\mathbf{K}_{uu}^{-1}\mathbf{K}_{u\hat{f}}(\boldsymbol{\theta}^*)), \tag{26}$$

and $p(\mathbf{u}) = \mathcal{N}(\mathbf{u}; \mathbf{0}, \mathbf{K}_{uu})$, using notation shortcuts $\mathbf{K}_{uu} := k(\mathbf{Z}_u, \mathbf{Z}_u)$, $\mathbf{K}_{\hat{f}u}(\boldsymbol{\theta}^*) := k(\mathbf{Z}(\boldsymbol{\theta}^*), \mathbf{Z}_u)$, and $\mathbf{K}_{\hat{f}\hat{f}}(\boldsymbol{\theta}^*) := k(\mathbf{Z}(\boldsymbol{\theta}^*), \mathbf{Z}(\boldsymbol{\theta}^*))$. We may now formulate an evidence lower bound (ELBO) based on the joint variational density $q(\hat{\mathbf{f}}, \mathbf{u}, \boldsymbol{\theta}^*)$ as:

$$
\begin{aligned}
\log p(\mathbf{y}) &= \mathbb{E}_{q(\hat{\mathbf{f}}, \mathbf{u}, \boldsymbol{\theta}^*)}\left[\log \frac{p(\mathbf{y}, \hat{\mathbf{f}}, \mathbf{u}, \boldsymbol{\theta}^*)}{q(\hat{\mathbf{f}}, \mathbf{u}, \boldsymbol{\theta}^*)}\right] + \mathbb{D}_{\mathrm{KL}}(q(\hat{\mathbf{f}}, \mathbf{u}, \boldsymbol{\theta}^*)||p(\hat{\mathbf{f}}, \mathbf{u}, \boldsymbol{\theta}^*|\mathbf{y})) \\
&\geq \mathbb{E}_{q(\hat{\mathbf{f}}, \mathbf{u}, \boldsymbol{\theta}^*)}\left[\log \frac{p(\mathbf{y}, \hat{\mathbf{f}}, \mathbf{u}, \boldsymbol{\theta}^*)}{q(\hat{\mathbf{f}}, \mathbf{u}, \boldsymbol{\theta}^*)}\right].
\end{aligned}
\tag{27}
$$

Since $\mathbb{D}_{\mathrm{KL}}(q(\hat{\mathbf{f}}, \mathbf{u}, \boldsymbol{\theta}^*)||p(\hat{\mathbf{f}}, \mathbf{u}, \boldsymbol{\theta}^*|\mathbf{y})) \geq 0$, and 0 if and only if $q(\hat{\mathbf{f}}, \mathbf{u}, \boldsymbol{\theta}^*) = p(\hat{\mathbf{f}}, \mathbf{u}, \boldsymbol{\theta}^*|\mathbf{y})$, maximising the ELBO above w.r.t. $q$ provides us with an approximation to the joint posterior. Choosing $q(\hat{\mathbf{f}}, \mathbf{u}, \boldsymbol{\theta}^*) := p(\hat{\mathbf{f}}|\mathbf{u}, \boldsymbol{\theta}^*)q(\mathbf{u}, \boldsymbol{\theta}^*)$ simplifies the ELBO to [53]:

$$\log p(\mathbf{y}) \geq \mathbb{E}_{q(\hat{\mathbf{f}}, \mathbf{u}, \boldsymbol{\theta}^*)}\left[\log \frac{p(\mathbf{y}|\hat{\mathbf{f}})p(\mathbf{u})p(\boldsymbol{\theta}^*)}{q(\mathbf{u}, \boldsymbol{\theta}^*)}\right]. \tag{28}$$

Sparse variational GP approaches can reduce the computational complexity of Bayesian inference on GPs to $\mathcal{O}(NM^2)$ or even $\mathcal{O}(M^3)$ [48, 54], where $N$ is the number of data points.

### B.2.2 Structure of the joint variational posterior

If we would take a mean-field approach setting $q(\mathbf{u}, \boldsymbol{\theta}^*) := q(\mathbf{u})q(\boldsymbol{\theta}^*)$, the ELBO above would further simplify, leading to a few computational advantages, as explored by Bayesian GP-LVM methods [53, 32, 54]. However, in our experimental design context, this approach leads to a few issues. Firstly, using the mean-field posterior as a replacement for our joint posterior breaks the dependence between $\hat{y}$ and $\boldsymbol{\theta}^*$, leading their mutual information (a.k.a. EIG) to be zero regardless of the design inputs $\hat{\mathbf{x}}$ and $\hat{\boldsymbol{\theta}}$. Secondly, although $\mathbf{u}$ and $\boldsymbol{\theta}^*$ are independent according to their priors (Eq. 25), they become dependent when conditioned on the data. In fact, the true posterior over $\mathbf{u}$ given the data and the true parameters $\boldsymbol{\theta}^*$ is exactly Gaussian:

$$p(\mathbf{u}|\mathcal{D}_t, \boldsymbol{\theta}^*) = \mathcal{N}(\mathbf{u}; \mu_t(\mathbf{Z}_u; \boldsymbol{\theta}^*), k_t(\mathbf{Z}_u, \mathbf{Z}_u; \boldsymbol{\theta}^*)), \tag{29}$$

where $\mu_t(\cdot; \boldsymbol{\theta}^*)$ and $k_t(\cdot, \cdot; \boldsymbol{\theta}^*)$ are given by Eq. 9 and Eq. 10, respectively. Note, however, that the posterior over $\boldsymbol{\theta}^*$ should not be Gaussian for a general non-linear kernel $k$. Therefore, it makes more sense for us to model $q(\mathbf{u}, \boldsymbol{\theta}^*) := q(\mathbf{u}|\boldsymbol{\theta}^*)q(\boldsymbol{\theta}^*)$. Moreover, learning a Gaussian conditional model over $\mathbf{u}$ and a flexible variational distribution over $\boldsymbol{\theta}^*$ should be enough to allow us to recover the true posterior, since $p(\mathbf{u}, \boldsymbol{\theta}^*|\mathcal{D}_t) = p(\mathbf{u}|\mathcal{D}_t, \boldsymbol{\theta}^*)p(\boldsymbol{\theta}^*|\mathcal{D}_t)$.

**Optimal variational inducing-point distribution.** Given $\boldsymbol{\theta}^* \in \Theta$, we have a standard sparse GP model. The optimal variational inducing-point distribution is available in closed form following standard results [48] as:

$$q^*(\mathbf{u}|\boldsymbol{\theta}^*) = \mathcal{N}(\mathbf{u}; \boldsymbol{\mu}_u(\boldsymbol{\theta}^*), \boldsymbol{\Sigma}_u(\boldsymbol{\theta}^*)), \tag{30}$$

where the distribution parameters are:

$$\boldsymbol{\mu}_u(\boldsymbol{\theta}) := \mathbf{K}_{uu}(\mathbf{K}_{uu} + \boldsymbol{\Psi}_2(\boldsymbol{\theta}))^{-1}\boldsymbol{\Psi}_1(\boldsymbol{\theta})^\top\mathbf{y} \tag{31}$$

$$\boldsymbol{\Sigma}_u(\boldsymbol{\theta}) := \mathbf{K}_{uu}(\mathbf{K}_{uu} + \boldsymbol{\Psi}_2(\boldsymbol{\theta}))^{-1}\mathbf{K}_{uu}, \tag{32}$$

and the conditional $\Psi$ matrices are given by:

$$\boldsymbol{\Psi}_1(\boldsymbol{\theta}) := \mathbf{K}_{\hat{f}u}(\boldsymbol{\theta})\boldsymbol{\Sigma}_{\mathbf{y}}^{-1} \tag{33}$$

$$\boldsymbol{\Psi}_2(\boldsymbol{\theta}) := \mathbf{K}_{u\hat{f}}(\boldsymbol{\theta})\boldsymbol{\Sigma}_{\mathbf{y}}^{-1}\mathbf{K}_{\hat{f}u}(\boldsymbol{\theta}), \tag{34}$$

for $\boldsymbol{\theta} \in \Theta$. The computational cost of sampling predictions with this model then reduces from $\mathcal{O}(N^3)$ to $\mathcal{O}(NM^2)$.

**Parametric variational inducing distribution.** To further reduce the computational cost of predictions, we may accept a sub-optimal conditional variational inducing-point distribution given by a parametric model:

$$q_{\boldsymbol{\zeta}}(\mathbf{u}|\boldsymbol{\theta}^*) := \mathcal{N}(\mathbf{u}; \mathbf{m}_{\boldsymbol{\zeta}}(\boldsymbol{\theta}^*), \boldsymbol{\Sigma}_{\boldsymbol{\zeta}}(\boldsymbol{\theta}^*)), \tag{35}$$

following the architecture in Sec. 5.3. This formulation allows us to approximate the evidence lower bound in Eq. 28 w.r.t. $q(\mathbf{u}|\boldsymbol{\theta}^*)$ via mini-batching [see 55]. To do so, we approximate $\hat{f}_i := \hat{f}(\mathbf{z}_i)$ via conditionally independent samples given $\mathbf{u}$, for $i \in \{1, \dots, N\}$. As a result, the data-dependent term in Eq. 28 decomposes as a sum which is amenable to mini-batching:

$$\mathbb{E}_{q_{\boldsymbol{\zeta}}(\hat{\mathbf{f}}, \mathbf{u}|\boldsymbol{\theta}^*)}[\log p(\mathbf{y}|\hat{\mathbf{f}})] \approx \sum_{i=1}^{N} \mathbb{E}_{q_{\boldsymbol{\zeta}}(\hat{f}_i, \mathbf{u}|\boldsymbol{\theta}^*)}[\log p(y_i|\hat{f}_i)] \tag{36}$$

where $q_{\boldsymbol{\zeta}}(\hat{f}_i, \mathbf{u}|\boldsymbol{\theta}^*) = p(\hat{f}_i|\mathbf{u}, \boldsymbol{\theta}^*)q_{\boldsymbol{\zeta}}(\mathbf{u}|\boldsymbol{\theta}^*)$. The variational parameters $\boldsymbol{\zeta}$ need to be optimised within a second optimisation loop after the data update in Algorithm 1 w.r.t.:

$$\ell_t(\boldsymbol{\zeta}) := \mathbb{E}_{q_t(\boldsymbol{\theta}^*)}\left[\sum_{i=1}^{N} \mathbb{E}_{q_{\boldsymbol{\zeta}}(\hat{f}(\mathbf{z}_i), \mathbf{u}|\boldsymbol{\theta}^*)}[\log p(y_i|\hat{f}(\mathbf{z}_i))]\right] - \mathbb{E}_{q_t(\boldsymbol{\theta}^*)}[\mathbb{D}_{\mathrm{KL}}(q_{\boldsymbol{\zeta}}(\mathbf{u}|\boldsymbol{\theta}^*)||p(\mathbf{u}))]. \tag{37}$$

Although the GP update is no longer available in closed form, we gain computational efficiency for large volumes of data. Applying mini-batches of size $L \ll N$ to Eq. 37 results in a computational cost $\mathcal{O}(LM^2)$ (or $\mathcal{O}(M^3)$, if $M > L$), which is smaller than the cost $\mathcal{O}(NM^2)$ of the optimal variational distribution $q^*(\mathbf{u}|\boldsymbol{\theta}^*)$.

## C Further discussion on limitations

**High-dimensional settings.** The dimensionality of our search space consists of the combined dimensionality of the designs $\mathcal{X}$ and calibration parameters space $\Theta$, which can be large in practical applications. In general, in higher dimensions, one is to expect that the algorithm will require a larger number of iterations to find suitable posterior approximations due to the possible increase in complexity of the posterior. The analysis of such complexity, however, is problem-dependent and outside the scope of this work. In addition, note that we do not mean that the per-iteration runtime is directly affected, since what dominates the cost of inference is sampling from the GP, whose runtime complexity is dominated by the cube of the number of data points due to a matrix inversion operation, while being only linear in dimensionality.

**Gaussian assumptions.** We make Gaussian assumptions when modelling the simulator and the approximation errors, which can be seen as restrictive for some applications. However, if the errors are sub-Gaussian (i.e., its tail probabilities decay faster than that of a Gaussian), as is the case for bounded errors, we conjecture that a GP model can still be a suitable surrogate, as it would not underestimate the error uncertainty. If the error function is sampled from some form of heavy-tailed stochastic process (e.g., a Student-T process), the GP would, however, tend to under estimate uncertainty and lead to possibly optimistic EIG estimates that make the algorithm under-explore the search space. Changing from a GP model to another type of stochastic process model that can capture heavier tails would be possible, though require significant changes to the algorithm's predictive equations. We, however, believe that most real-world cases would present errors which are at least bounded (and therefore sub-Gaussian) with respect to the simulations.

