# OpenReview forum: "Bayesian Adaptive Calibration and Optimal Design"
_NeurIPS.cc/2024/Conference — NeurIPS 2024 poster_

### Official Review · Reviewer_CjgP · 2024-07-10

**Soundness:** 4
**Presentation:** 3
**Contribution:** 3
**Rating:** 7
**Confidence:** 3

**Summary:**

This paper addresses the problem of calibrating simulation models. Simulation models depend on inputs set by the user, referred to as designs, and parameters representing unknown physical quantities, called calibration parameters. The task is to find calibration parameters such that simulations match real observations. To that end, the authors propose an active learning scheme in which maximally informative design and calibration parameters are iteratively chosen to construct the training set. They also propose a Gaussian process structure adapted to this setting.

**Strengths:**

### Originality
* This piece of work is new to me.

### Quality
* The method is sound, and I did not identify any flaws.
* Related works are discussed.
* The experiments support the claims.

### Clarity
* The paper is well articulated.
* The problem is clearly introduced, and the intuition behind the proposed solution is provided early on.
* Figures are clean.

### Significance
* Although my knowledge of the field is too limited to have a strong opinion, the problem addressed seems important.

**Weaknesses:**

### Originality
* I have no concerns regarding originality.

### Quality
* I have no concerns regarding the quality.

### Clarity
* Calibration parameters are also referred to as simulation parameters, which is confusing.
* Some minor comments for the camera-ready version. There is a typo in equation 6 (missing parenthesis). In algorithm 1, in the "update posterior" line, I believe this should be $\mathcal{D}_t$. Table 3 is not referred to in section 6.4.

### Significance
* I have no concerns regarding significance.

**Questions:**

I have no questions.

**Limitations:**

I do not see any unaddressed limitations.

---

> ### Author Rebuttal · Authors · 2024-08-07
>
> We would like to thank the reviewer for their positive feedback and for letting us know about the clarity issues. Table 3 consists of the results discussed in Sec. 6.4, which unfortunately missed a direct reference to the table. We will make sure to address these and the other issues in the revision.

---

> > ### Comment · Reviewer_CjgP · 2024-08-12
> >
> > Thanks for the update.
> >
> > I have read the rebuttal and the other reviews and keep my score unchanged.

---

### Official Review · Reviewer_bPdt · 2024-07-12

**Soundness:** 3
**Presentation:** 3
**Contribution:** 2
**Rating:** 3
**Confidence:** 4

**Summary:**

The paper proposes a more data-efficient algorithm inspired by Bayesian adaptive experimental design. This algorithm runs maximally informative simulations in a batch-sequential process, estimating posterior distribution parameters and optimal designs by maximizing a variational lower bound of the expected information gain. The algorithm is validated on both synthetic and real-data problems.

**Strengths:**

1. The paper is well-written, offering comprehensive background information and a thorough review of the literature. The method is rigorously compared with other related approaches across multiple metrics on both synthetic and real-data datasets.
2. The paper focuses on a well-motivated and challenging problem.

**Weaknesses:**

The paper can be improved in the following ways:
1. My biggest concern for this paper is its novelty: replacing EIG by a variational lower bound has been explored and well studied by many literature. (e.g. [18], [29])
2. The method is currently compared only against "Random" and "IMSPE". However, there exist numerous other variants of Bayesian optimal design and frequentist approaches with diverse optimality criteria. It would be valuable to compare the proposed method with these state-of-the-art alternatives.
3. The variational inference is usually applied to sampling from posterior with large-scale datasets or high-dimensional parameter spaces. But the paper only present results on small datasets with low-dimensional parameter spaces, which makes it less convincing.

**Questions:**

1. In Figure 1:  (a) It's challenging to ascertain if stability has been achieved, which affects the credibility of the conclusion that BACON achieves rapid convergence in terms of MAP estimates. (b). It's difficult to determine if EIG statistically outperforms others in terms of RMSE.
2. The paper lacks clarity in specifying its contributions and novelty. Could you please elaborate on the main distinction between BACON and other Bayesian optimal design methods that incorporate variational inference (e.g. Variational Bayesian Optimal Experimental Design)?
3. There are numerous instances where variational inference (VI) can fail and produce poor approximations of the target posterior distribution. It would be beneficial to investigate the performance of BACON in scenarios where VI struggles to capture the characteristics of the target distribution, such as when dealing with highly correlated coordinates or multimodal targets.

**Limitations:**

Variational inference (VI) is typically used to approximate posterior distributions in large-scale datasets and high-dimensional parameter spaces. However, the paper indicates that BACON, which uses VI in Bayesian optimal design, still struggles to scale to large datasets. This raises the question of why VI is used in BACON instead of traditional MCMC sampling methods like HMC. The rationale for choosing VI over these traditional methods remains unclear.

---

> ### Author Rebuttal · Authors · 2024-08-07
>
> We would like to thank the reviewer for the detailed comments and insightful feedback. We provide a global response clarifying the contrast with the state-of-the-art and details on an additional baseline, but further elaborate on specific details relevant to the reviewer's comments below.
>
> **Novelty.** Concerning novelty, we highlight that it is not trivial to apply existing Bayesian adaptive experimental design (BAED) methods to the calibration setting with an expensive black-box simulator. Existing adaptive experimental design approaches (e.g., Foster et al. [13, 16]) rely on \emph{unrestricted} access to the simulator $h$ to generate simulated observations $\hat{y}$, while in our case we have a limited budget due to the cost of simulations (e.g., long runtime). An alternative would be to sample observations from the GP model, instead, and perhaps learn an adaptive design policy based on GP-based observation histories [16, 17]. The GP then would serve as a proxy for the simulator and the actual simulations would serve as the ``real'' experiments within a typical BAED framework. However, sampling from a GP breaks important conditional independence assumptions which these methods rely on, since simulation outcomes $\hat{y}_1, \dots, \hat{y}_t$ sampled from a GP would no longer be conditionally independent given the true calibration parameters $\boldsymbol{\theta}^*$, as $p(\hat{y}_t | \boldsymbol{\theta}^*, \mathbf{\hat{x}}_t, \boldsymbol{\hat{\theta}}_t)$ $\neq$ $p(\hat{y}_t | \boldsymbol{\theta}^*,  \mathbf{\hat{x}}_t, \boldsymbol{\hat{\theta}}_t, \mathcal{D}\_{t-1})$
> under a GP prior for a latent simulator $h$. In addition, the most appropriate prior for these methods would not be the unconditional prior $p(\boldsymbol{\theta}^*)$, but $p(\boldsymbol{\theta}^*|\mathbf{y}_R)$, instead, which is non-trivial.
>
> **Contrast with the state of the art.** Besides the reasons outlined above, one could think of Bayesian optimal experimental design methods that consider black-box simulators [14, 19]. However, these approaches often only consider the calibration space [19], keeping the designs fixed, or are focused on the problem of finding a single optimal design for a real experiment [14]. We chose VBMC as a representative for approaches which focus on the choice of calibration parameters, while keeping the design points fixed. VBMC's performance is presented in Table 1 to 3 in the manuscript in addition to the other two baselines.
>
> **Additional baseline.** For the rebuttal, we are including experimental results with an additional baseline representing a classic experimental design criterion. We implemented a strategy which selects designs of maximal predictive entropy for a GP model with only the simulation data. Such strategy can be related to the D-optimality criterion when considering the stochastic process representing the unknown simulator as the random variable of interest. We include a table with the new results in the PDF as part of our author rebuttal.In additional, we also show above that the IMSPE is A-optimal and equivalent to the Active Learning Cohn (ALC) criterion used in other active learning methods (Sauer et al., 2022, in Author Rebuttal above). Our new results show that BACON is still superior in terms of maximising the expected information gain (EIG), as measured by the KL divergence between the final (MCMC) posterior estimate $p_T$ and the true posterior $p^*$ (with perfect knowledge of the simulator).
>
> **Reason for variational inference.** We chose a variational approximation as it allows us to formulate a lower bound on the expected information gain (EIG) and perform *joint optimisation and inference* with a single well-defined objective. This is perhaps more difficult to achieve through sampling.  VI for us is then simply a framework to reformulate both inference and design problems as optimisation. Despite the similarities, however, note that our formulation works by minimising the *forward* KL divergence $KL(p||q)$, instead of the usual reverse $KL(q||p)$ which VI works with. The forward KL tends to match the moments of the target distribution better, while reverse KL mainly focuses on the modes. As such, it would behave differently in the situations which the reviewer described. On multimodal targets, for example, an optimal forward-KL approximation would tend to spread its mass across the detected modes. Therefore, even if the approximation is limited due to the representation power of the chosen parametric family (i.e., conditional normalising flows in our case), the uncertainty on the parameters estimate can still be captured, which is the main driver for an information-theoretic criterion, like the EIG. In addition, a reverse KL objective could lead to certain pathologies that cause discontinuities in the EIG estimation, due to the variational posterior possibly locking to very different modes (see Appendix G in Foster et al. [13]), which we avoid with the forward KL.
>
> **Performance in Figure 1.** A large part of the variance of the results in Figure 1 and Table 1 is actually due to the variability of the GP samples used as simulators for each individual run. The average behaviour of the metrics (as presented by the solid mean curves), therefore, is more indicative of the actual performance of the algorithms. In that case, we see that the MAP error drops quicker for BACON than for the other algorithms, as well as the RMSE values.

---

> > ### Comment · Reviewer_bPdt · 2024-08-13
> >
> > Thank you for your response. However, I still find Figure 1 unclear. The confidence intervals of the different methods overlap, suggesting that the proposed method may not be significantly better than the alternatives. As a result, my concerns remain, and I would like to retain my current score.

---

> ### Author Response · Authors · 2024-08-14
>
> Thanks for the feedback. We understand the reviewer's concern, as we perhaps have not selected the most appropriate performance metrics to show, and the discussion is lacking a few key insights. We would like to point out that most of the posteriors in the synthetic calibration problem of Sec. 6.2 are multimodal, since the simulators are simply random functions drawn from a GP prior and only 5 "real" data points were provided, alongside 20 initial simulation points. In these multimodal problems, MAP estimates, which show most of the confidence interval overlaps in Fig. 1, should not be considered a primary metric of performance, since the mode of the posterior will often not match the true calibration parameter, given the low amount of data.  We, however, have been able to show that our proposed method (BACON) achieves its intended goal, which is to maximise the EIG, as measured by the expected KL divergence from final to initial posterior $\mathbb{D}_{\mathrm{KL}}(p_T||p_0)$ (see Eq. 1 for the equivalence), when compared to the baselines in the paper across all experimental benchmarks. We will clarify these points in the revision and include a few examples of some of the posterior distributions we find in the synthetic and real data problems to better illustrate such challenges.

---

### Official Review · Reviewer_Z11Z · 2024-07-12

**Soundness:** 3
**Presentation:** 3
**Contribution:** 2
**Rating:** 7
**Confidence:** 3

**Summary:**

This paper addresses the challenge of calibrating expensive-to-evaluate computer models using Bayesian adaptive experimental design. The novelty of the proposed method (BACON) lies in using the expected information gain (EIG), which is a principled information theoretic criterion for active learning, to perform calibration of models. Another point of departure from the existing literature is the fact that BACON performs active learning in the joint space of design and parameters.

**Strengths:**

This is a technically solid and well-written paper which I really enjoyed reading. The idea of using the EIG criterion over the joint space of $\theta$ and $x$ may sound simple and straightforward once you see it written down that it is almost surprising that no has done this before.  Many good papers seem obvious once you read them, and I feel like this paper belongs in that category. Despite being a notation heavy paper, the authors did a good job of making the writing clear and concise.

**Weaknesses:**

I do not have any major concerns. Some questions/comments that might help improve the paper are as follows:
* It would have been nice to see an ablation study where the design and parameters are not jointly optimized over, in order to ascertain the benefit of doing so.
* The error function $\varepsilon$ is also modellled as a GP. I wonder how does BACON's performance get affected if this assumption does not hold (i.e. when this error model is misspecified).
* Perhaps a discussion of the hyperparameters/settings of the proposed method would be nice to have, in terms of how to set them and how sensitive the performance of BACON is to their values.
* MMD is not mentioned in the text despite being plotted in Figure 1(d).
* Table 3 is not referenced in Section 6.4.
* Please include up/down arrows in the tables next to the columns so that it is easy to read them.
* Appendix B seems incomplete.

**Questions:**

* Can you say something about the tightness of the bound in Section 5.1? What does it depend on? (I suppose these would have been discussed in ref [13] but it would be nice if mentioned here as well)
* Can you explain a bit more why VBMC performs better in the synthetic experiments but fares poorly in the other experiments? Does it have anything to do with the dimensionality of the problem?
* Reporting the KL divergence between the prior and the posterior after T iterations tells us how much the posterior has changed, but that does not mean we are converging to the true posterior, right (we may be confidently biased)? Is the KL divergence between $p_T$ and $p^*$ a better indicator for accuracy?

**Limitations:**

The authors have discussed the limitations adequately.

---

> ### Author Rebuttal · Authors · 2024-08-07
>
> We would like to thank the reviewer for their positive and insightful feedback. We will revise the text addressing the issues raised. In addition to our global response in the main author rebuttal, our response to the reviewer's specific questions follows below.
>
> ### Weaknesses
> Regarding the issues raised as weaknesses, we elaborate on answers below.
> > It would have been nice to see an ablation study where the design and parameters are not jointly optimized over, in order to ascertain the benefit of doing so.
>
> To optimise only the calibration parameters while keeping the designs fixed usually involves setting them to the designs in the real data. However, the choice of designs to use for each chosen calibration parameter in the batch, especially when the batch size and number of real data points do not match, leads to some ambiguities and not a very obvious pathway to implementation. Our current  proxy for the non-joint-optimisation case is VBMC, whose performance is presented in Table 1 to 3.
>
> > The error function $\epsilon$ is also modelled as a GP. I wonder how does BACON's performance get affected if this assumption does not hold (i.e. when this error model is misspecified).
>
> If the errors are still sub-Gaussian (i.e., its tail probabilities decay faster than that of a Gaussian), as is the case for bounded errors, we conjecture that a GP model can still be a suitable surrogate, as it would not underestimate the error uncertainty. If the error function is sampled from some form of heavy-tailed stochastic process (e.g., a Student-T process), the GP would, however, tend to under estimate uncertainty and lead to possibly optimistic EIG estimates that make the algorithm under-explore the search space. Changing from a GP model to another type of stochastic process model that can capture heavier tails would be possible, though require significant changes to the algorithm's predictive equations. We, however, believe that most real-world cases would present errors which are at least bounded (and therefore sub-Gaussian) with respect to the simulations. We will add a discussion to the revision.
>
> >Perhaps a discussion of the hyperparameters/settings of the proposed method would be nice to have, in terms of how to set them and how sensitive the performance of BACON is to their values.
>
> Besides the GP hyperparameters (e.g., lengthscales, noise variance, etc.), there are optimisation settings (i.e., step sizes, scheduling rates, etc.), conditional density model hyper-parameters (i.e., normalising flow architecture), and algorithm's settings, e.g., the designs batch size $B$. The latter is dependent on the available computing resources (e.g., number of CPU cores or compute nodes for simulations in a high-performance computing system). We tuned optimisation settings and architectural parameters for the conditional normalising flows via Bayesian optimisation with short runs (e.g., 10-30 iterations) on the synthetic problem. However, depending on the number of parameters, a simpler approach, like grid search, might be enough. The GP hyper-parameters were optimised via maximum a posteriori estimation after each iteration's batch update. We will elaborate on the details of our hyper-parameter optimisation setup in the revision.
>
> Thanks for noticing the other minor issues, which we will make sure to address in the revised text. MMD stands for maximum mean discrepancy, an integral probability metric that quantifies the distance between two probability distributions based on their embeddings in a reproducing kernel Hilbert space. We used the implementation available in the ITE package [39]. We will add a proper definition and reference in the revision.
>
> ### Answers to questions
> **Tightness of the variational EIG bound.**
> The bound in Sec. 5.1, quantifying the difference between the variational EIG and the true EIG, is exact. A full derivation can be found in Foster et al. [13, Appendix A.1], but we will add it to the revision, as suggested.
>
> **VBMC's performance** VBMC relies on a mixture of Gaussians as the variational approximation to the posterior, which tends to be quite smooth. Therefore, the algorithm may struggle to approximate non-smooth posteriors, as is the case for the location finding problem, for example. Another potential point of struggle for VBMC is in problems where the posterior's mode is not concentrated, but spread along a path, e.g., a circle, as shown in Oliveira et al. [22]. The dimensionality of the calibration parameters space may also affect VBMC's posterior approximation, since Gaussian components can be seen as a form of radial basis functions, whose representation power deteriorates with the increase in dimensionality. Another key difference to note is that VBMC does not produce an emulator approximating the simulator, but an approximation to the log-likelihood function, in contrast to the other GP-emulation-based methods we assess.
>
> **KL divergence performance criteria.** The expected information gain is equivalent to the expected KL divergence between the posterior and the prior (Eq. 1) and quantifies the expected reduction in uncertainty (i.e., entropy) on the posterior after incorporating observations produced by a given set of designs. The average KL in the tables, therefore, can be seen as an approximation to the final EIG. In our setup, we are approximating a latent function representing the simulator, and as the number of observations grows to infinity, that approximation should asymptotically concentrate at the true simulator. The posterior obtained with the true simulator is simply $p^*$. However, there might be a long gap between an optimal EIG and its asymptotic limit, which might not be possible to quantify. Computing the KL divergence between the final posterior approximation $p_T$ and $p^*$ is then a more direct measure of how close the algorithm has got to the asymptotic posterior.

---

> > ### Comment · Reviewer_Z11Z · 2024-08-09
> >
> > I thank the authors for their clarifying responses to my questions. The general response outlining the difference between experimental design and Bayesian calibration makes the utility of their approach clearer. I think this paper is a useful contribution when calibrating parameters of a computationally expensive mechanistic model that depends on a design variable (which is slightly different to the kind of models calibrated using simulation-based inference). Hence, I am happy to recommend accept.

---

### Official Review · Reviewer_bRKR · 2024-07-12

**Soundness:** 2
**Presentation:** 2
**Contribution:** 2
**Rating:** 5
**Confidence:** 4

**Summary:**

This paper considers the problem of calibration of computer models as an active learning problem. Given the objective being maximizing the expected information gain (EIG) about calibration parameters, and based on the assumption of linear dependency between simulator outcome and true observation, this work proposes a Gaussian process model that jointly models true observations, calibration parameters and simulator outcomes, and maximization of EIG is performed based on this model. Due to the intractability of the EIG objective, the authors further propose to use variational objective in replace of the original EIG in finding the next design parameters and calibration parameters to sample.

**Strengths:**

- The problem setting is interesting and is indeed important in engineering and physical sciences as computer simulators are often used in those areas.
- The approach of using a single GP to model calibration parameters and design parameters jointly is novel.

**Weaknesses:**

- Optimizing calibration parameters and design parameters jointly seems to make the problem harder because dimension of the search space is the sum of both spaces, therefore, the applicability of this method to real-world problems may be limited.

**Questions:**

- The proposed approach certainly makes sense when both spaces of calibration parameters and design parameters are low-dimensional. It may be helpful to include a discussion about choosing appropriate method when the number of calibration parameters is bigger or the number of design parameters is bigger, or both.

**Limitations:**

Please also include limitation of the dimensionality issue as pointed out above.

---

> ### Author Rebuttal · Authors · 2024-08-07
>
> We would like to thank the reviewer for their comments and feedback. Indeed, the dimensionality of the problem consists of the sum of the dimensionalities of the design space and the calibration parameters space. The purpose of this paper, however, was to propose a general method for Bayesian calibration via adaptive experimental design and some of the aforementioned limitations were out of our current scope. In high-dimensional settings, there are two main issues to address.
>
> * GP models with conventional translation-invariant radial kernels start to lose predictive performance in high-dimensional spaces due to the disappearing volume of the $n$-ball at higher dimensions.
> * Finding optimal design points in higher dimensions may become increasingly difficult due to the vast spaces.
>
> The first issue may be addressed by incorporating more specialised, possibly non-stationary, kernels, which better capture correlations in higher-dimensional spaces. For example, Li et al. (2024, below) have recently shown that infinite-width Bayesian neural networks (BNNs) maintain satisfying modelling capabilities in higher-dimensional active learning settings. Due to their equivalence with GPs (Lee et al., 2018, below), infinite-width BNNs can be implemented as a GP model with a kernel given by the infinite-width limit covariance function of a BNN of a given architecture. A few results are known in the literature and software packages are available for such. The main point, however, is that this approach would not lead to any modifications to our proposed algorithm, since it is still GP-based. We will make these clarifications along with the corresponding set of reference in the revised manuscript.
>
> The second issue may result in slower, though still computationally feasible, convergence of optimisation approaches based on stochastic gradient descent (see Arous et al, 2021, below). To mitigate possible issues with finding good local optima, multiple random restarts can be applied.
>
> In general, in higher dimensions, one is to expect that the algorithm will require a larger number of iterations $T$ to find suitable posterior approximations due to the possible increase in complexity of the posterior $p(\boldsymbol{\theta}^*|\mathcal{D}_t)$. The analysis of such complexity, however, is problem dependent and outside the scope of this work. In addition, note that we do not mean that the per-iteration runtime is directly affected, since what dominates the cost of inference is sampling from the GP, whose runtime complexity $\mathcal{O}(t^3 + dt^2)$ is dominated by the cube of the number of data points due to a matrix inversion operation, while being only linear in dimensionality $d$ (the term $dt^2$ is the cost of computing the $t^2$ entries in the kernel matrix $\mathbf{K}_t$). We will expand on these limitations in the final version of the paper.
>
> **References**
> * Arous, G. Ben, Gheissari, R., and Jagannath, A. (2021). Online stochastic gradient descent on non-convex losses from high-dimensional inference. *Journal of Machine Learning Research*, 22.
>  * Lee, J., Bahri, Y., Novak, R., Schoenholz, S. S., Pennington, J., and Sohl-dickstein, J. (2018). *Deep neural networks as Gaussian processes*. In International Conference on Learning Representations (ICLR).
>  * Li, Y. L., Rudner, T. G. J., and Wilson, A. G. (2024). *A Study of Bayesian Neural Network Surrogates for Bayesian Optimization*. 2024 International Conference on Learning Representations (ICLR 2024).

---

> > ### Comment · Reviewer_bRKR · 2024-08-13
> >
> > Thanks for the response. I have read other comments as well and decide to keep my scoring unchanged.

---

### Author Rebuttal · Authors · 2024-08-07

We would like to thank all the reviewers for their constructive feedback and the time and effort applied in reviewing our manuscript. We provide individual responses to each review, but we also address some of the main common points here. In addition, we have *new results* with additional baselines available in the PDF attached to this rebuttal.

# Applicability of existing BAED methods
In contrast to traditional Bayesian adaptive experimental design (BAED) approaches, the Bayesian calibration problem presents a few key differences and challenges.
* **The experiment *is* the simulation.** Our problem is to select informative simulation designs $(\mathbf{\hat{x}}, \boldsymbol{\hat{\theta}})$ to run in order to reduce uncertainty about calibration parameters $\boldsymbol{\theta}^*$ (e.g., physical properties) that influence real observed data. The possibly expensive simulations, therefore, are the experiments we run, and the role of the real data is to form our prior beliefs on the unknown parameters of interest, which are then updated using simulations.
* **A model for the simulator (a.k.a. an emulator) is needed.** To run state-of-the-art BAED approaches, we would need an emulator for the simulator that is able to correlate the simulation outcomes with the unknown parameter $\boldsymbol{\theta}^*$ for the real data. In our case, we use a GP model, which is a natural choice for Bayesian emulation, given its well known capabilities and guarantees. However, a GP also models correlations across the outcomes, which breaks the conditional independence assumption across simulation outcomes (e.g., $\hat{y}_i, \hat{y}_j$), since: $$p(\hat{y}_i, \hat{y}_j|\boldsymbol{\theta}^*, \mathbf{y}_R) = p(\hat{y}_i| \hat{y}_j, \boldsymbol{\theta}^*, \mathbf{y}_R) p(\hat{y}_j|\boldsymbol{\theta}^*, \mathbf{y}_R)\neq p(\hat{y}_i|\boldsymbol{\theta}^*, \mathbf{y}_R)p(\hat{y}_j|\boldsymbol{\theta}^*, \mathbf{y}_R),$$ an important assumption which existing methods rely on. Ignoring it would lead to an over-estimation of the predictive variances of $\hat{y}$ by using the GP predictive marginal $p(\hat{y}_i|\boldsymbol{\theta}^*, \mathbf{y}_R)$, instead of the conditional $p(\hat{y}_i| \hat{y}_j, \boldsymbol{\theta}^*, \mathbf{y}_R)$, besides also shifting the predictive mean, leading to biases in the EIG and misguiding the algorithm.
* **The "prior" is non-trivial.** The prior $p(\boldsymbol{\theta}^*|\mathbf{y}_R)$ for the calibration problem is actually a posterior over the parameters $\boldsymbol{\theta}^*$ given the real data $\mathbf{y}_R$. In contrast, standard BAED approaches assume the prior is relatively simple to sample from and evaluate densities of, possibly with a closed-form expression. So we also cannot rely on easily evaluating prior densities or on sampling from them, except for approximations provided by, e.g., MCMC or VI.

Considering the points above, the application of existing BAED methods to our setting is not straightforward and led us to the development of a dedicated algorithmic framework to tackle these issues.

# Additional baseline
We provide experimental results with an additional baseline following a *D-optimality* criterion, a classic experimental design objective, in the rebuttal PDF. Optimal candidate designs according to this criterion are points of maximum uncertainty according to the model [28]. If we model the simulator $h$ as the unknown variable of interest, this corresponds to selecting designs where we have maximum entropy of the Gaussian predictive distribution $p(\hat{y}|\mathbf{\hat{x}}, \boldsymbol{\hat{\theta}}, \mathbf{\hat{y}}_t)$. This approach, therefore, simply attempts to collect an informative set of simulations according to the GP prior only, without considering the real data. Running D-optimality on $\boldsymbol{\theta}^*$, in contrast, would lead back to the EIG criterion we use.

We would also like to highlight that the IMSPE criterion, which we use as a baseline, is equivalent to the Active Learning Cohn (ALC) criterion (Sauer et al., 2022, below) and a form of *A-optimality* [28], another classic experimental design objective. The ALC maximises the reduction in predictive variance of the model across a set of reference designs. Since the current predictive variance is constant w.r.t. the new data, maximising the ALC is equivalent to minimising the IMSPE. In addition, an integral over the predictive variance is also evaluating the trace of the posterior covariance operator of the simulations GP. Namely, letting $\Sigma := \mathbb{E}[h \otimes h ]$ denote the covariance operator of $h\sim \mathcal{GP}(0, k)$, assuming $h$ is realised in an $L_2(\Xi)$ space of square-integrable functions of $\xi := (\mathbf{\hat{x}}, \boldsymbol{\hat{\theta}}) \in \Xi := \mathcal{X} \times \Theta$, we have that:
$$\mathrm{Tr}(\Sigma) = \mathrm{Tr}(\mathbb{E}[h \otimes h ]) = \mathbb{E}[\mathrm{Tr}(h \otimes h) ] = \mathbb{E}[\lVert h \rVert_2] = \mathbb{E}\left[\int_{\Xi} h(\xi)^2 d\xi \right] = \int_{\Xi} \mathbb{E}\left[ h(\xi)^2 \right] d\xi = \int_{\Xi} \sigma^2(\xi) d\xi,$$
where we applied the linearity of the trace and of the expectation. A-optimality seeks designs $\xi$ which maximally reduce the trace of the posterior covariance of the unknown variable, i.e., maximising the difference $\mathrm{Tr}(\Sigma) - \mathbb{E}[\mathrm{Tr}(\Sigma)|\xi, \hat{y}]$. Therefore, maximising the ALC (or minimising the IMSPE) is equivalent to selecting A-optimal points. We revised our IMSPE implementation based on the methodology presented in Sauer et al. (2022) and included the results in the PDF.

Our new results show that BACON can still reach superior performance when compared to classic experimental design criteria in terms of expected information gain (EIG), while keeping competitive performance in terms of posterior approximation.

**References**

* Sauer, A., Gramacy, R. B., \& Higdon, D. (2022). Active Learning for Deep Gaussian Process Surrogates. *Technometrics, 65*(1).

---

### Decision · Program_Chairs · 2024-09-25

**Decision:**

Accept (poster)

**Comment:**

This paper introduces a method for calibrating expensive computer models while jointly optimizing design parameters using Bayesian adaptive experimental design. The approach employs expected information gain for calibration and performs active learning in the joint space of design and parameters.
The reviewers generally agree that the paper addresses an important problem in engineering and physical sciences. Two reviewers recommend acceptance. One reviewer gives a borderline accept, while another recommends rejection. In the final discussion, however, reviewers highlighted the paper as technically solid and well-written as well as novel.
The authors are encouraged to incorporate their rebuttal explanations and additional results into the final version of the paper.